# Learning in Games with Lossy Feedback

**Zhengyuan Zhou**
Stanford University
zyzhou@stanford.edu

**Panayotis Mertikopoulos**
Univ. Grenoble Alpes, CNRS, Inria, LIG
panayotis.mertikopoulos@imag.fr

**Susan Athey**
Stanford University
athey@stanford.edu

**Nicholas Bambos**
Stanford University
bambos@stanford.edu

**Peter Glynn**
Stanford University
glynn@stanford.edu

**Yinyu Ye**
Stanford University
yinyu-ye@stanford.edu

## Abstract

We consider a game-theoretical multi-agent learning problem where the feedback information can be lost during the learning process and rewards are given by a broad class of games known as variationally stable games. We propose a simple variant of the classical online gradient descent algorithm, called reweighted online gradient descent (ROGD) and show that in variationally stable games, if each agent adopts ROGD, then almost sure convergence to the set of Nash equilibria is guaranteed, even when the feedback loss is asynchronous and arbitrarily corrrelated among agents. We then extend the framework to deal with unknown feedback loss probabilities by using an estimator (constructed from past data) in its replacement. Finally, we further extend the framework to accomodate both asynchronous loss and stochastic rewards and establish that multi-agent ROGD learning still converges to the set of Nash equilibria in such settings. Together, these results contribute to the broad lanscape of multi-agent online learning by significantly relaxing the feedback information that is required to achieve desirable outcomes.

## 1 Introduction

In multi-agent online learning [13, 14, 21, 45], a set of agents repeatedly interact with the environment and each other while accumulating rewards in an online manner. The key feature in this problem is that to each agent, the environment consists of all other agents who are simultaneously making such sequential decisions, and hence, each agent's reward depends not only on their own action, but also on the joint action of all other agents. A common way to model reward structures in this multi-agent online learning setting is through a repeated but otherwise *unknown* stage game: the reward of the $i$-th agent at iteration $n$ is $r_i(a_i^n, a_{-i}^n)$, where $a_i^n$ is agent $i$'s action at $n$ and $a_{-i}^n$ is the vector of all other agents's actions at stage $n$.

Even though the underlying stage game is fixed (i.e. each $r_i(\cdot)$ is fixed), from each agent's own perspective, its reward is nevertheless a time-varying function when viewed solely as a function of its own action, i.e., $r_i^n(\cdot) = r_i(\cdot, a_{-i}^n)$, and it needs to select an action $a_i^n$ before receiving the reward $r_i^n(a_i^n)$. As such, each agent is exactly engaged in a classical online learning process [9, 23, 42, 43]. In this context, there is a fruitful line of existing literature providing a rich source of online learning algorithms to minimize the standard performance metric known as *regret* [10], defined as $\text{Reg}_T^{\textbf{Alg}} = \max_{a_i \in \mathcal{A}} \sum_{t=1}^{T} \{r_i^t(a_i) - r_i^t(a_i^t)\}$ with the sequence of actions $a_i^t$ generated by some online learning algorithm **Alg**. These algorithms, already classical, include "follow-the-regularized-leader" [26], online gradient descent [57], multiplicative/exponential weights [3], online mirror descent [44], and many others. Perhaps the simplest algorithm in the above list is Zinkevich's online gradient descent (OGD) method, where the agent simply takes a gradient step (using their

current reward function) to form the action for the next stage, performing a projection if the iterate steps out of the feasible action set. This algorithm, straightforward as it is, provides (asymptotically) optimal regret guarantees (see Section B in the appendix for a brief review) and is arguably one of the most well-studied and widely-used algorithms in online learning theory and applications [24, 39, 57].

Consequently, several natural questions arise in multi-agent online learning : if each agent adopts OGD to minimize regret, what is the resulting evolution of the joint action? Specifically, under what games/assumptions would it lead to a Nash equilibrium (the leading solution concept in multi-agent games)? These questions fall in the broader inquiry of understanding the joint convergence of no-regret online learning algorithms, an inquiry that lies at the heart of game-theoretical learning for well-grounded reasons: Specifically, studying whether the process converges at all provides an answer as to the stability of the joint learning dynamics, while studying whether it converges to a Nash equilibrium provides an answer as to whether the joint learning dynamics lead to the emergence of rationality. Specifically, for the latter point, if, when agents adopt an online learning algorithm the joint action converges to a point that is not a Nash equilibrium, each agent would be able to do better by unilaterally deviating from that algorithm. Hence, convergence to a non-Nash point would inherently produce "regret" in equilibrium.

## 1.1 Related Work

Despite the fact that game-theoretic learning has received significant scrutiny in the literature, the questions raised above are still open for several reasons.

**First**, in general, joint convergence of no-regret learning does not hold. In fact, even in simple finite games (where each agent has a finite number of actions), no-regret learning may fail to converge [31]. Even if it does converge, in mixed extensions of finite games (where each agent's action set is a probability simplex over a finite number of actions), the limit can assign positive weight only to strictly dominated actions. Consequently, establishing joint convergence to Nash equilibria under no-regret learning algorithms (OGD included) for a broad and meaningful subclass of games has been a challenging ongoing effort.

**Second**, the extensive existing literature in the field has mainly focused on studying convergence in (mixed extensions of) finite games [8, 34, 45, 46]. Earlier work of game-theoretic learning (see [18] for a comprehensive review) has mainly focused on learning in finite games with dynamics that are not necessarily regret-less (e.g. best-response dynamics, fictitious play and the like). Subsequently, the seminal work [14] (see also the many references therein) has provided a unified treatment of joint convergence of various no-regret learning algorithms in mixed extensions of finite games. The primary focus of [14] is convergence to other, coarser equilibrium notions (e.g. correlated equilibria), where a fairly complete characterization is given. On the other hand, as pointed out in [14], convergence to Nash is a much more difficult problem: recent results of [47] have clearly highlighted the gap between (coarse) correlated equilibria obtained by no-regret learning processes and Nash equilibria. More positive results can be obtained in the class of potential games where, in a recent paper, the authors of [15] established the convergence of multiplicative weights and other regularized strategies in potential games with only payoff-based, bandit feedback.

However, moving beyond mixed extensions of finite games, much less is known, and only a moderate amount of literature exists. In the context of mixing in games with continuous action spaces, the authors of [37] provide a convergence analysis for a perturbed version of the multiplicative weights algorithm in potential games. In a pure-strategy setting, the network games considered in [29] belong to the much broader class of games known as concave games: each agent's reward function is individually concave.[1] Therein, the dynamics investigated may lead to positive regret in the limit. The recent paper [5] studied a two-player continuous zero-sum game, and showed that if both players adopt a no-regret learning algorithm, then the empirical time-average of the joint action converges to Nash equilibria. However, barring a few recent exceptions, the territory of no-regret learning on concave games is not well understood (let alone in general games with continuous action sets). An exception to this are the recent papers [30, 32] where the authors establish the convergence of mirror

descent in monotone games – a result later extended to learning with bandit, zeroth-order feedback in [7, 11].

**Third**, the convergence mode that is commonly adopted in the existing literature is ergodic convergence (i.e. convergence of $\frac{1}{n}\sum_{t=n}^{t} a^t$), rather than the convergence of the actual sequence of play (i.e. $a^n$). The former is convergence of the empirical frequency of play, while the latter is convergence of actual play: note that the latter implies the former, but not the other way round. It is important to point out, however, that convergence of the actual sequence of play is more desirable[2] *not only* because it is theoretically more appealing, but also because it characterizes the joint evolution of the system in no-regret learning (since $a^n$, rather than $\frac{1}{n}\sum_{t=n}^{t} a^t$, is the *actual* action taken).

This issue was highlighted in [31], where it is shown that even though continuous-time follow-the-regularized-leader (another no-regret learning algorithm) converges to Nash equilibrium in linear zero-sum games in the sense of time-averages, actual play orbits Nash equilibria in perpetuity. However, most of the existing work focus on establishing ergodic convergence (which granted is some form of stability), where the tools developed are far from sufficient to ensure convergence of actual play. Some recent exceptions in specific games do exist: [27, 28] studied nonatomic routing games (a special class of concave games) and established that both online mirror descent and multiplicative weights (yet another two no-regret algorithms) converge to Nash equilibria; while [36] established multiplicative weights converge to Nash equilibria under certain conditions in atomic routing games (a subclass of finite games). Theoretically, the main difficulty in establishing convergence of the sequence of play lies in cleverly designing (as done in [27, 28, 36, 53]) specific and much sharper Lyapunov functions for the specific learning algorithms at hand. This difficulty is partially overcome in [11] for strongly monotone games where, assuming perfect gradient observations, an $\mathcal{O}(1/n)$ convergence rate is established.

## 1.2 Our Contributions

In this paper, we study the convergence of OGD to Nash equilibrium in general continuous games. To the best of our knowledge, the existing state-of-the-art convergence guarantee is that multi-agent OGD converges to Nash equilibria in *monotone* games in the sense of **ergodic averages**, a result due to [35]. Monotone games[3] are an important and well-known subclass of concave games (in particular, it includes convex potential games as a special case). On a related note, a very recent paper [41] considered pseudo-monotone games (which strictly contain monotone games but also belong to concave games) and devised two distributed algorithms for computing Nash equilibria. However, it is far from clear that the algorithms devised are no-regret when used in an online fashion.

In this paper, we work with a broad class of general continuous games (not necessarily concave even), known as variationally stable games [53, 54], that strictly include pseudo-monotone games (and hence, in a chain of inclusion, monotone games and convex potential games), and also nonatomic routing games, linear Cournot games, resource allocation auctions, etc. More importantly, we go a step further and consider an even more general multi-agent online learning problem, where we allow agents to have asynchronous gradient feedback losses (note that so far the game-theoretical learning discussion is under *perfect* feedback, where the resulting landscape is already challenging). More specifically, instead of assuming that every agent receives information at every stage, we allow for cases where a (random) subset of agents remains informed (specifically, each agent $i$ has a probability $p_i$ of receiving the feedback and probability $1 - p_i$ of losing it).

Two important features of this model are: 1) the feedback loss can be asynchronous; 2) the feedback loss can be *arbitrarily correlated* among agents (whether some agent loses its feedback on the current iteration may affect whether others lose theirs). In this asynchronous context, we design a simple variant of OGD, called *reweighted online gradient descent* (ROGD), where each agent corrects its own marginal bias via dividing the received gradient (if any) by $p_i$. This is inspired by the classical EXP3 algorithm [4, 12]) in the multi-armed literature, where it has a similar feature of weighting the observed bandit feedback by the probability of selecting the corresponding arm. We then establish in

Theorem 4.3 that, in this asynchronous feedback loss setting, the sequence of play induced by joint ROGD converges *almost surely* to Nash equilibria in variationally stable games. We achieve this strong theoretical guarantee by designing an energy function that sharply tracks the progress made in the joint evolution of the ROGD update of all agents under feedback loss. We mention that a very recent work that also studies multi-agent online learning under imperfect information at the generality of variationally stable games is [53]. In particular, it is shown there that online mirror descent (also no-regret) converges to Nash even in the presence of super-linear but sub-quradtic growing delays in gradients. However, in that context, in addition to studying a different algorithm, [53] focuses on delays and in particular requires all gradients to be received (i.e. no gradient loss is allowed). Consequently, the results in [53] are strictly complementary to ours here: in particular, it is unclear whether online mirror descent would converge to Nash under feedback loss.

Finally, we make two practically useful and theoretically meaningful extensions. **First**, we extend to the case where the loss probabilites (which are assumed to be known so far) are not known. In this case, we can replace the actual probabilities $p_i$ by an estimate. Our main message (Theorem 5.1) is that provided the estimator is $\sqrt{n}$-consistent, then convergence to Nash in last iterate under ROGD is still guaranteed. Note that a simple estimator that satisfies this guarantee is the sample mean estimator (with Laplace smoothing). **Second**, we extend the multi-agent online learning setup to also accomodate for stochastic reward functions, where in each iteration only a random reward function is realized. In such cases, we first note an important equivalence result, both from a modeling standpoint and from a proof standpoint: the setup where agents' reward functions are *iid* can be identified with the setup where agents' reward functions are fixed and deterministic but the received feedback gradients are noisy (but unbiased). We then establish that (in either setup) multi-agent ROGD learning still converges almost surely to the set of Nash equilibria when noise has bounded support (Theorem E.4 in the appendix); and converges in probability to the set of Nash equilibria if noise has unbounded support but has finite second moment (Theorem E.6 in the appendix): both are in last iterate. Due to space limitation, this part is placed in the appendix. Together, these results not only make meaningful progress towards the challenging open problem of convergence of no-regret algorithms to Nash in general continuous games under perfect information, but also, more importantly, contribute to the broad lanscape of multi-agent online learning under imperfect information.

## 2 Problem setup

### 2.1 Rewards for Multi-Agent Online Learning

We consider a general continuous game model for the rewards in multi-agent online learning:

**Definition 2.1.** A *continuous game* $\mathcal{G}$ is a tuple $\mathcal{G} = (\mathcal{N}, \mathcal{A} = \prod_{i=1}^{N} \mathcal{A}_i, \{r_i\}_{i=1}^{N})$, where $\mathcal{N}$ is the set of $N$ *agents*, $\mathcal{X}_i$ is a convex and compact subset of $\mathbb{R}^{d_i}$ representing the *action space* of agent $i$, and $r_i \colon \mathcal{A} \to \mathbb{R}$ is agent $i$'s *reward function*, such that $r_i(a) = r_i(a_1, a_2, \ldots, a_N)$ is continuous in $a$ and continuously differentiable in $a_i$ for each $i \in \mathcal{N}$ and $\nabla_{a_i} r_i(a)$ is Lipschitz continuous in $a$.

**Definition 2.2.** Given a continuous game $\mathcal{G}$, $a^* \in \mathcal{X}$ is called a Nash equilibrium if for each $i \in \mathcal{N}$, $r_i(a_i^*, a_{-i}^*) \geq r_i(a_i, a_{-i}^*), \forall a_i \in \mathcal{X}_i$.

For the rest of the paper, we set $d = \sum_{i=1}^{N} d_i$, which denotes the dimension of the joint action space: $\mathcal{A} \subset \mathbb{R}^d$. Variables associated with agent $i$ are denoted by subscripts. If there is no subscript, then it is understood that it is a joint variable of all agents. When each agent is applying[4] OGD independently, we obtain multi-agent OGD in Algorithm 1 (we assume $\sum_{n=1}^{\infty} \gamma_{n+1} = \infty, \sum_{n=1}^{\infty} \gamma_{n+1}^2 < \infty$):

**Algorithm 1:** Multi-Agent OGD Learning

---

**Require:** Each agent $i$ picks an arbitrary $a_i^0 \in \mathbb{R}^{d_i}$

 1: $n \leftarrow 0, y_i^0 \leftarrow a_i^0$
 2: **repeat**
 3:    **for** each agent $i$ **do**
 4:       $y_i^{n+1} = y_i^n + \gamma_{n+1} \nabla_{a_i} r_i(a^n)$
 5:       $a_i^{n+1} = \mathbf{Proj}_{\mathcal{A}_i}(y_i^{n+1})$
 6:    **end for**
 7:    $n \leftarrow n + 1$
 8: **until** end

---

## 2.2 Asynchronous Feedback Loss

The setup described in Algorithm 1 is multi-agent learning under perfect feedback information. Here we extend the model to allow for asynchronous feedback loss, where at each time step, only a subset $\mathcal{N}^{n+1} \subset \mathcal{N}$ of the agents receive their gradients from time[5] $n$, while other agents' gradients are lost. We assume that this subset is drawn **iid** from a fixed distribution across time steps.

To faciliate the discussion, let the indicator variable $I_i^{n+1}$ be 1 if agent $i$'s gradient from $n$ is received (at the beginning of $n + 1$) and 0 otherwise. Mathematically:

$$I_i^{n+1} = \begin{cases} 1, & \text{if } i \in \mathcal{N}^{n+1}, \\ 0, & \text{if } i \notin \mathcal{N}^{n+1}. \end{cases} \tag{2.1}$$

Note that under this model, even though $\mathcal{N}^{n+1}$ is drawn **iid** across time steps, within the same time step, $I_i^{n+1}$ and $I_j^{n+1}$ can be arbitrarily correlated (whether agent $i$'s feedback is lost can influence whether agent $j$'s feedback is lost). We assume each agent $i$ only knows its own marginal loss probability $1 - p_i$, where $p_i = \mathbb{E}[I_i^{n+1}]$. We refer to this setup as asynchronous feedback loss, because different agents can have different feedback loss pattern at any iteration. Applying multi-agent OGD in the feedback loss model yields:

---

**Algorithm 2:** Multi-Agent OGD Learning under Asynchronous Feedback Loss

---

**Require:** Each agent $i$ picks an arbitrary $A_i^0 \in \mathbb{R}^{d_i}$

 1: $n \leftarrow 0, Y_i^0 \leftarrow A_i^0$
 2: **repeat**
 3:    **for** each agent $i$ **do**
 4:       $Y_i^{n+1} = \begin{cases} Y_i^n + \gamma_{n+1} \nabla_{a_i} r_i(A^n), & \text{if } I_i^{n+1} = 1 \\ Y_i^n, & \text{if } I_i^{n+1} = 0 \end{cases}$
 5:       $A_i^{n+1} = \mathbf{Proj}_{\mathcal{A}_i}(Y_i^{n+1})$
 6:    **end for**
 7:    $n \leftarrow n + 1$
 8: **until** end

---

Here we have capitalized the iterates $Y_i^n, A_i^n$ because they are now random variables (due to random feedback loss). Specifically, denoting by $I$ the vector of indicator variables for all the agents, we have that both $Y_i^n$ and $A_i^n$ are adapted to $A^0, I^1, I^2, \ldots, I^n$. Note the important but subtle point here: $Y_i^n$ and $A_i^n$ are **not** adapted to $A_i^0, I_i^1, I_i^2, \ldots, I_i^n$, because each individual's gradient update is coupled with all the other agents' actions, which is a fundamental phenomenon in multi-agent learning.

## 2.3 Variationally Stable Games

Consequently, special structures must be imposed on the games/reward functions. Here we consider a broad *meta* class of games, called variationally stable games [32, 54], that contain many existing

well-known classes of games, such as convex potential games, monotone games, pseudo-monotone games [56], coercive games [41], influence network games [52, 55], non-atomic routing games [40]. See [53] for more details.

**Definition 2.3.** $\mathcal{G}$ is a variationally stable game if its set of Nash equilibria $\mathcal{A}^*$ is non-empty and satisfies: $\langle \nabla_a r(a), a - a^* \rangle \triangleq \sum_{i=1}^{N} \langle \nabla_{a_i} r_i(a), a_i - a_i^* \rangle \leq 0, \forall a \in \mathcal{A}, \forall a^* \in \mathcal{A}^*$, with equality if and only if $a \in \mathcal{A}^*$.

*Remark* 1. Monotone games require $\langle \nabla_a r(a) - \nabla_a r(a'), a - a' \rangle \leq 0, \forall a, a' \in \mathcal{A}$. Pseudo-monotone games require: if $\langle \nabla_a r(a'), a - a' \rangle \leq 0$, then $\langle \nabla_a r(a), a - a' \rangle \leq 0, \forall a, a' \in \mathcal{A}$. That monotone is a special case of pseudo-monotone is obvious. That pseduo-monotone is a special case of variational stability follows by recalling the standard characterization of a Nash equilibrium: $a^*$ satisfies $\langle \nabla_a r(a^*), a - a^* \rangle \leq 0, \forall a \in \mathcal{A}$.

In the presence of feedback loss, the convergence of multi-agent OGD given in Algorithm 2 to Nash equilibria cannot be guaranteed in variationally stable games (and there exist cases where it doesn't, in fact, convergence of OGD is not guaranteed even in convex potential games). In the next section, we present a simple modification of the vanilla multi-agent OGD that will later be shown to converge to Nash equilibria, even in the presence of asynchronous delays. We shall then see that as a corollary (see Corollay 4.4), multi-agent OGD will converge to the set of Nash equilibria (almost surely) if the feedback loss are synchronous (and as a further special case, if there is no feedback loss).

# 3 Algorithm and energy function

In this section, to deal with asynchronous feedback loss, we give a new algorithm called Reweighted Online Gradient Descent (ROGD) for the multi-agent learning problem.

## 3.1 Reweighted Online Gradient Descent

The main idea in the modification of OGD lies in each agent individually correcting its marginal bias (which comes from the loss of the gradient feedback) by dividing the probability. More specifically, when a gradient is lost on the current time step, the agent uses the previous action as in OGD. Otherwise, when a gradient is indeed available, the gradient will be weighted by $p_i$ first before getting updated. This results in reweighted online gradient descent:

---

**Algorithm 3:** Multi-Agent ROGD Learning under Asynchronous Feedback Loss

---

**Require:** Each agent $i$ picks an arbitrary $A_i^0 \in \mathbb{R}^{d_i}$
1: $n \leftarrow 0, Y_i^0 \leftarrow A_i^0$
2: **repeat**
3:     **for** each agent $i$ **do**
4:         $Y_i^{n+1} = \begin{cases} Y_i^n + \gamma_{n+1} \frac{\nabla_{a_i} r_i(A^n)}{p_i}, & \text{if } I_i^{n+1} = 1 \\ Y_i^n, & \text{if } I_i^{n+1} = 0 \end{cases}$
5:         $A_i^{n+1} = \mathbf{Proj}_{\mathcal{A}_i}(Y_i^{n+1})$
6:     **end for**
7:     $n \leftarrow n + 1$
8: **until** end

---

We emphasize once again that the gradient loss among agents can be correlated. However, in ROGD, each agent only corrects its own marginal bias and does not concern itself with other agents' feedback loss. To analyze the ROGD algorithm, we next introduce an important theoretical tool.

## 3.2 Energy Function

We start by defining the energy function, an important tool we use to establish convergence later. $a^*$ is always any fixed Nash equilibrium.

**Definition 3.1.** Define the energy function $L : \mathbf{R}^d \to \mathbf{R}$ as follows:

$$L(y) = \|a^*\|_2^2 - \|\mathbf{Proj}_{\mathcal{A}}(y)\|_2^2 + 2\langle y, \mathbf{Proj}_{\mathcal{A}}(y) - a^* \rangle. \tag{3.1}$$

**Lemma 3.2.** *Let $y^n$ be a sequence in $\mathbb{R}^d$*

1. $L(y^n) \to 0$ *implies* $\mathbf{Proj}_\mathcal{A}(y^n) \to a^*$.

2. $\|\mathbf{Proj}_\mathcal{A}(y) - \hat{y}\|_2^2 - \|\mathbf{Proj}_\mathcal{A}(\hat{y}) - \hat{y}\|_2^2 \le \|y - \hat{y}\|_2^2$, *for any* $y, \hat{y} \in \mathbb{R}^d$

3. $L(y + \Delta y) - L(y) \le 2\langle \Delta y, \mathbf{Proj}_\mathcal{A}(y) - a^* \rangle + \|\Delta y\|_2^2$, *for any* $y, \Delta y \in \mathbb{R}^d$.

*Remark* 2. The second statement of the lemma serves as an important intermediate step in proving the third statement, and is established by leveraging the envelop theorem and several important properties of Euclidean projection. To see that this is not trivial, consider the quantity $\|\mathbf{Proj}_\mathcal{A}(y) - \hat{y}\|_2 - \|\mathbf{Proj}_\mathcal{A}(\hat{y}) - \hat{y}\|_2$, which we know by triangle's inequality satisfies:

$$\|\mathbf{Proj}_\mathcal{A}(y) - \hat{y}\|_2 - \|\mathbf{Proj}_\mathcal{A}(\hat{y}) - \hat{y}\|_2 \le \|\mathbf{Proj}_\mathcal{A}(y) - \mathbf{Proj}_\mathcal{A}(\hat{y})\|_2 \le \|y - \hat{y}\|_2, \quad (3.2)$$

where the last inequality follows from the fact that projection is an non-expansive map. However, this inequality is not sufficient for our purposes because in quantifying the perturbation $L(y + \Delta y) - L(y)$, we also need the squared term $\|\Delta y\|_2^2$, which is not easily obtainable from Equation (3.2). In fact, a finer-grained analysis is needed to establish that $\|y - \hat{y}\|_2^2$ is an upper bound on $\|\mathbf{Proj}_\mathcal{A}(y) - \hat{y}\|_2^2 - \|\mathbf{Proj}_\mathcal{A}(\hat{y}) - \hat{y}\|_2^2$.

# 4 Almost sure convergence to Nash equilibria

In this section, we establish that when agents' rewards come from a variationally stable game, multi-agent ROGD converges to the set of Nash equilibria almost surely under asynchronous feedback loss. For ease of exposition, we break the framework into three steps, each of which centers on one idea and is described in detail in a subsection. All the proof details are given in the appendix. For the first two subsections, let $a^*$ be an arbitrary Nash equilibrium.

## 4.1 Controlling the Tail Behavior of Expectation

Our first step lies in controlling the tail behavior of the expected value of the sequence $\{\langle \nabla_a r(A^n), a^* - A^n \rangle\}_{n=0}^\infty$. Note that by variational stability, we have:

$$\forall n, \langle \nabla_a r(A^n), a^* - A^n \rangle \ge 0, \text{ a.s.}$$

Consequently, $\mathbb{E}[\langle \nabla_a r(A^n), a^* - A^n \rangle] \ge 0, \forall n$. By leveraging the energy function, its telescoping sum and an appropriate conditioning, we show that (next lemma) this sequece of expectations should be rather small in the limit and its tail should "mostly" go to 0.

**Lemma 4.1.**

$$\sum_{t=0}^\infty \gamma_{n+1} \mathbb{E}[\langle \nabla_a r(A^t), a^* - A^t \rangle] < \infty. \quad (4.1)$$

*Remark* 3. Since $\sum_{t=0}^\infty \gamma_{n+1} = \infty$, Lemma 4.1 that $\liminf_{n \to \infty} \mathbb{E}[\langle \nabla_a r(A^t), a^* - A^t \rangle] = 0$. Note that the converse is not true: when a subsequence of $\{\mathbb{E}[\langle \nabla_a r(A^n), a^* - A^n \rangle]\}_{n=0}^\infty$ converges to 0, the sum need not be finite. As a simple example, consider $\gamma_{n+1} = \frac{1}{n}$, and

$$\mathbb{E}[\langle \nabla_a r(A^n), a^* - A^n \rangle] = \begin{cases} \frac{1}{n}, \text{if } t = 2^k \\ 1, \text{otherwise.} \end{cases} \quad (4.2)$$

Then the subsequence on indicies $2^k$ converges to 0, but the sum still diverges. This means that Equation (4.1) is stronger than subsequence convergence.

## 4.2 Bounding the Successive Differences

However, Equation (4.1) is still not strong enough to guarantee that $\lim_{n \to \infty} \mathbb{E}[\langle \nabla_a r(A^t), a^* - A^t \rangle] = 0$, let alone $\lim_{n \to \infty} \langle \nabla_a r(A^t), a^* - A^t \rangle = 0$, a.s.. This is because the convergent sum given in Equation (4.1) only limits the tail growth somewhat, but not completely. As an example to demonstrate this point, let $\mathcal{C}_t$ be the following boolean variable[6]:

$$\mathcal{C}_t = \begin{cases} 1, \text{if } t \text{ contains the digit } 9 \text{ in its decimal expansion} \\ 0, \text{otherwise.} \end{cases} \quad (4.3)$$

Now define $\gamma_{n+1} = \frac{1}{n}$, and $\mathbb{E}[\langle \nabla_a r(A^n), a^* - A^n \rangle] = \begin{cases} \frac{1}{n}, \text{if } \mathcal{C}_t = 1 \\ 1, \text{if } \mathcal{C}_t = 0. \end{cases}$ Then it follows from a straightforward calculation that $\sum_{t=0}^{\infty} \gamma_{n+1} \mathbb{E}[\langle \nabla_a r(A^t), a^* - A^t \rangle] < \infty$ (see Problem 1.3.24 in [16]). However, the limit $\mathbb{E}[\langle \nabla_a r(A^t), a^* - A^t \rangle]$ does not exist.

This indicates that to obtain almost sure convergence of $\langle \nabla_a r(A^n), a^* - A^n \rangle$ to 0, we need to impose more stringent conditions to ensure its sufficient tail decay. One way is to bound the difference between every two successive terms in terms of a decreasing sequence. This ensures that $\langle \nabla_a r(A^t), a^* - A^t \rangle$ cannot change two much from iteration to iteration. Further, the change between two successive terms will become smaller and smaller. This result is formalized in the following lemma (the proof is given in the appendix):

**Lemma 4.2.** *There exists a constant $C > 0$ such that for every $n$,*

$$\langle \nabla_a r(A^{n+1}), a^* - A^{n+1} \rangle - \langle \nabla_a r(A^n), a^* - A^n \rangle \leq C\alpha_{n+1}, \textbf{\textit{a.s.}} \tag{4.4}$$

## 4.3 Main Convergence Result

We are now finally ready to put all the pieces together and state the main convergence result.

**Theorem 4.3.** *Let the reward functions be given from a variationally stable game. Define the point-to-set distance in the standard way: $\text{dist}(a, \mathcal{A}^*) = \inf_{a^* \in \mathcal{A}^*} \|a - a^*\|_2$. Then for any strictly positive probabilities $\{p_i\}_{i=1}^N$, ROGD converges almost surely to the set of Nash equilibria: $\lim_{n \to \infty} \text{dist}(A^n, \mathcal{A}^*) = 0$ a.s., as $n \to \infty$, where $A^n$ is a sequence generated from Algorithm 3.*

*Remark* 4. As a quick outline here (the details are in appendix), pick an arbitrary Nash equilibrium $a^* \in \mathcal{A}^*$. Lemma 4.1 and Lemma 4.2 will together ensure that $\lim_{n \to \infty} \langle \nabla_a r(A^t), a^* - A^t \rangle = 0$, a.s. Since $\langle \nabla_a r(a), a^* - a \rangle > 0$ if and only if $a \notin \mathcal{A}^*$ and $\langle \nabla_a r(a), a^* - a \rangle = 0$ if and only if $a \in \mathcal{A}^*$, it then follows by continuity of $\nabla_a r(\cdot)$ that $\lim_{n \to \infty} \langle \nabla_a r(A^t), a^* - A^t \rangle = 0$, a.s. implies $\lim_{n \to \infty} \text{dist}(A^n, \mathcal{A}^*) = 0$ a.s..

Although not mentioned in this theorem, another useful and interesting structural insight to point out here is that as $\langle \nabla_a r(A^t), a^* - A^t \rangle$ converges to 0, if it ever becomes 0 at $n$, then $A^t \in \mathcal{A}^*$, and furthermore, the joint action will stay **exactly** at that Nash equilibrium forever. Why? There are two cases to consider.

1. First, this Nash equilibrium $a^*$ is an interior point in $\mathcal{A}$. In this case, $\nabla_a r(a^*) = 0$ and hence $\nabla_a r(A^n) = 0$. Consequently, per the ROGD update rule, whether any agent updates or not does not matter: either the gradient is not received, in which case no gradient update happens; or a gradient is received, but at this Nash equilibrium, it is 0 and therefore nobody will want to make any update.

2. Second, this Nash equilibrium $a^*$ is a boundary point in $\mathcal{A}$. In this case, $\nabla_a r(a^*)$ may not be 0, but it always points outside the feasible action set $\mathcal{A}$. Consequently, even if an agent receives a gradient and hence makes a gradient update, its action will immediately get projected back to the same point. As a result, the joint action $A^n$ will stay exactly at $\mathcal{A}$ (even though the $Y^n$ variables will still change).

**Corollary 4.4.** *Under the same setup as in Theorem 4.3, if feedback loss is synchronous on average $(p_i = p_j, \forall i, j)$, then multi-agent OGD in Algorithm 2 converges almost surely to $\mathcal{A}^*$ in last iterate.*

This can be easily seen that by noting that the probability can be absorbed into the step-size.

**Corollary 4.5.** *Under the same setup as in Theorem 4.3, if there is no feedback loss $(p_i = 1, \forall i)$, multi-agent OGD in Algorithm 2 converges to $\mathcal{A}^*$ in last iterate.*

*Remark* 5. Note that as stated, our above results require a joint learning step-size policy, even if there is no missing gradient feedback – otherwise, the players' individual gradients weighted by individual step-sizes might not be "stable" and can cause divergence (and, of course, the situation only becomes worse in the "lossy" regime). That said, our analysis still holds if each player $i \in \mathcal{N}$ uses an individual step-size policy $\gamma_n^i$ such that $\limsup_{n \to \infty} \gamma_n^i / \gamma_n^j < \infty$ for all pairs $i, j \in \mathcal{N}$, i.e., if the players' updates do not follow different "time-scales" (so to speak). In that case however, the proofs (and the overall write-up) would become much more cumbersome, so we avoided this extra degree of generality in this paper.

# 5 Extension: Unknown Loss Probabilities

So far we have assumed $p_i$'s are known. We close the paper with a brief comment on how to remove this assumption. When each agent $i$ does not know the underlying loss probability $p_i$, ROGD is no longer feasible. To overcome this, we use an estimator $\hat{p}_i$ (obtainable from the past history) in replacement of the true probability $p_i$. Since the only information an agent has is the past history of received gradients, we require the estimator $\hat{p}_i$ to be adapted to the sequence of the indicator functions $I_i^t$'s: $\hat{p}_i^n = \hat{p}_i^t(I_i^1, \ldots, I_i^n)$. The resulting algorithm will be called EROGD. An estimator $\hat{p}^n$ is called $\sqrt{n}$-consistent if $\mathbf{E}[(\hat{p}^n - p)^2] = O(\frac{1}{n})$, where $p$ is the true parameter (note that it is called $\sqrt{n}$-consistent because root-mean-squared error is typically used to define the rate of consistency.). We have the following result (proof omitted due to space limitation):

**Theorem 5.1.** *Under the same setup as in Theorem 4.3, if $\hat{p}_i^n$ is $\sqrt{n}$-consistent for every $i \in \mathcal{N}$, and if $\sum_{n=1}^{\infty} \gamma_n \frac{1}{\sqrt{n}} < \infty$, then the last iterate of EROGD converges to $\mathcal{A}^*$ in probability.*

*Remark* 6. One simple estimator that is $\sqrt{n}$-consistent is sample mean with smoothing (where the smoothing is used to prevent the estimator from ever reaching 0): $p_i^{n+1} = \frac{\sum_{s=1}^{n} I_i^s + 1}{n+1}$. Further, many step-size sequences satisfy the requirement: examples include $\gamma_n = \frac{1}{n^\alpha}$ where $0.5 < \gamma < 1$.

# 6 Concluding remarks

In this paper, we have provided an algorithmic framework to deal with multi-agent online learning under feedback loss and obtained broad convergence-to-Nash results under fairly general settings. We also believe more exciting work remains. For instance, our formulation is game-theoretical where participating agents are self-interested. A parallel facet to multi-agent online learning is coordination [1, 2, 17, 20], where participating agents coordinate to achieve a common goal. Understanding how to effectively cooperate under imperfect information will be an interesting future direction. Another direction is to incorporate state into the reward and allow actions to also depend on the underlying state that may transition. Such settings belong broadly to multi-agent online policy learning, where the imperfect information regime is under-explored. Empirically, we believe the recent advances in deep learning and representation learning could possibly [19, 33, 48–50] provide a flexible architecture for learning good policies in the imperfect information regime, although characterizations of theoretical guarantees may require novel machinery. Finally, it would also be interesting to further extend the results into partial feedback settings. In the presence of a single agent, such problems have been studied in the context of offline policy learning [6, 51] and online bandits (with imperfect information) [22, 25, 38]. However, the multi-agent learning setting is again under-explored; we leave that for future work.

## Footnotes

[1]In this vein, mixed extensions of finite games can be called linear games, because each agent's reward is individually linear in its own action. Note also that convex potential games is a subclass of concave games. The following gives the set membership: finite games $\subset$ mixed extension of finite games $\subset$ concave games $\subset$ general continuous games.

[2]Of course, convergence in ergodic average is still valuable in many contexts, particularly when convergence of actual play fails to hold.

[3]In short, it means $(\nabla_{a_1} r_1(\cdot), \nabla_{a_2} r_2(\cdot), \ldots, \nabla_{a_n} r_n(\cdot))$ is a monotone operator on the joint action space. Note that if there is only 1 player, this means the underlying problem is a convex optimization problem since the gradient of a convex function is monotone. In this case, a Nash equilibrium is a globally optimal solution.

[4]Note that in particular, the gradient here is a partial gradient with respect to one's own action, and its dimension is equal to $d_i$, the dimension of its own action space.

[5] Here the superscript $n + 1$ means that set $\mathcal{N}^{n+1}$ is revealed at the beginning of time $n + 1$

[6]By it definition, $c_9 = 1, c_{11} = 0$.

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
