[Supplementary Material · Supplement.pdf]

# Supplementary Material for Learning in Games with Lossy Feedback

**Zhengyuan Zhou**
Stanford University
zyzhou@stanford.edu

**Panayotis Mertikopoulos**
Univ. Grenoble Alpes, CNRS, Inria, LIG
panayotis.mertikopoulos@imag.fr

**Susan Athey**
Stanford University
athey@stanford.edu

**Nicholas Bambos**
Stanford University
bambos@stanford.edu

**Peter Glynn**
Stanford University
glynn@stanford.edu

**Yinyu Ye**
Stanford University
yinyu-ye@stanford.edu

## A    Auxiliary results

We state four useful auxiliary results that will be used later.

The first one is from [2].

**Lemma A.1.** *Let $\mathcal{X}$ be a compact and convex subset of $\mathbb{R}^d$. Then for any $a \in \mathcal{X}, y \in \mathbb{R}^d$:*

$$\langle \boldsymbol{Proj}_{\mathcal{X}}(y) - a, \boldsymbol{Proj}_{\mathcal{X}}(y) - y \rangle \leq 0.$$

The second one is the envelope theorem [3].

**Lemma A.2.** *Let $f : \mathbb{R}^n \times \mathbb{R}^m \to \mathbb{R}$ be a continuously differentiable function. Let $U$ be a compact set and consider the problem*

$$\max_{x \in U} f(x, \theta). \tag{A.1}$$

*Let $x^* : \mathcal{O} \to \mathbb{R}^m$ be a continuous function defined on an open set $\mathcal{O} \subset \mathbb{R}^m$ such that for each $\theta \in \mathcal{O}$, $x^*(\theta)$ solves the problem in Equation A.1. Define $V : \mathbb{R}^m \to \mathbb{R}$ where $V(\theta) = f(x^*(\theta), \theta)$. Then $V(\theta)$ is differentiable on $\mathcal{O}$ and:*

$$\nabla V(\theta) = \nabla f(x^*(\theta), \theta). \tag{A.2}$$

The next two concern behavior of (deterministic and random) sequences.

The third one can be found in [1].

**Lemma A.3.** *Let $a_n, b_n$ be two non-negative sequences such that $\sum_{n=1}^{\infty} a_n = \infty, \sum_{n=1}^{\infty} a_n b_n < \infty$. If there exists a $K > 0$ such that $|b_{n+1} - b_n| \leq K a_n$. Then, $\lim_{n \to \infty} b_n = 0$.*

The last one is Lemma A6 from [5].

**Lemma A.4.** *Let $\{X_n\}_{n=1}^{\infty}$ be a sequence of non-negative random variables on a probability space. Let $\{\gamma_n\}_{n=1}^{\infty}, \{\beta_n\}_{n=1}^{\infty}$ be two non-negative sequences such that $\sum_{n=1}^{\infty} \gamma_n = \infty$ and $\sum_{n=1}^{\infty} \gamma_n \beta_n < \infty$. Assume there exists a constant $C$ such that for all $n \geq 1$, $\mathbb{E}[X_n] \leq \beta_n$ and $|X_{n+1} - X_n| \leq C\gamma_n$ almost surely. Then $X_n \to 0, a.s..$*

## B    Problem setup

### B.1    Prelude: Single-Agent Online Learning

We start with a brief recap on the standard single-agent online learning problem [6]. In single-agent online learning, an agent has an action space $\mathcal{A} \subset \mathbb{R}^d$ that is a convex and compact set and is

interacting with an unknown environment. On each iteration $n$, the agent takes an action $a^t$, observes the reward function $r_t(\cdot)$ (that can be time-varying) and recevies the reward $r_t(a^t)$. The agent aims to select actions so as to minimize regret $\text{Reg}_T^{\textbf{Alg}} = \max_{a \in \mathcal{A}} \sum_{t=1}^{T} \{r_t(a) - r_t(a^t)\}$, where the actions $a^t$'s are generated by the learning algorithm **Alg**.

A classical result therein is that provided $r_t(\cdot)$'s are concave functions and the gradient $\nabla r_t(a_t)$ is available on each iteration[1], a simple algorithm, known as online gradient descent[2] (OGD) achieves $O(\sqrt{T})$ regret that is universally tight [7]. Futher, online gradient descent (OGD) achieves $O(\log T)$ regret if $r_t(\cdot)$'s are strongly concave with bounded first and second derivatives [4]. In OGD (see Algorithm 1), $\textbf{Proj}_{\mathcal{X}}(\cdot)$ is the Euclidean projection operator: it projects the iterate to the closest point (in Euclidean distance) in the action space (if it ever steps out of $\mathcal{X}$). $\gamma_n$ is a step-size sequence that satisfies the usual summability condition: $\sum_{n=1}^{\infty} \gamma_n^2 < \infty, \sum_{n=1}^{\infty} \gamma_n = \infty$.

---
**Algorithm 1:** Online Gradient Descent Learning

---
**Require:** An arbitrary $a^0 \in \mathbb{R}^d$
1: $n \leftarrow 0, y^0 \leftarrow a^0$
2: **repeat**
3:     $y^{n+1} = y^n + \gamma_{n+1} \nabla r_t(a^n)$
4:     $a^{n+1} = \textbf{Proj}_{\mathcal{X}}(y^{n+1})$
5:     $n \leftarrow n + 1$
6: **until** end

---

We next list all the algorithms mentioned in the main text for ease of use, as they will be referenced throughout the results and proofs in this appendix.

---
**Algorithm 2:** Multi-Agent OGD Learning

---
**Require:** Each agent $i$ picks an arbitrary $a_i^0 \in \mathbb{R}^{d_i}$
1: $n \leftarrow 0, y_i^0 \leftarrow a_i^0$
2: **repeat**
3:     **for** each agent $i$ **do**
4:        $y_i^{n+1} = y_i^n + \gamma_{n+1} \nabla_{a_i} r_i(a^n)$
5:        $a_i^{n+1} = \textbf{Proj}_{\mathcal{X}_i}(y_i^{n+1})$
6:     **end for**
7:     $n \leftarrow n + 1$
8: **until** end

---

---
**Algorithm 3:** Multi-Agent OGD Learning under Asynchronous Feedback Loss

---
**Require:** Each agent $i$ picks an arbitrary $A_i^0 \in \mathbb{R}^{d_i}$
1: $n \leftarrow 0, Y_i^0 \leftarrow A_i^0$
2: **repeat**
3:     **for** each agent $i$ **do**
4:        $Y_i^{n+1} = \begin{cases} Y_i^n + \gamma_{n+1} \nabla_{a_i} r_i(A^n), & \text{if } I_i^{n+1} = 1 \\ Y_i^n, & \text{if } I_i^{n+1} = 0 \end{cases}$
5:        $A_i^{n+1} = \textbf{Proj}_{\mathcal{X}_i}(Y_i^{n+1})$
6:     **end for**
7:     $n \leftarrow n + 1$
8: **until** end

**Algorithm 4:** Multi-Agent ROGD Learning under Asynchronous Feedback Loss
---
**Require:** Each agent $i$ picks an arbitrary $A_i^0 \in \mathbb{R}^{d_i}$
1: $n \leftarrow 0, Y_i^0 \leftarrow A_i^0$
2: **repeat**
3:     **for** each agent $i$ **do**
4:       $Y_i^{n+1} = \begin{cases} Y_i^n + \gamma_{n+1} \frac{\nabla_{a_i} r_i(A^n)}{p_i}, & \text{if } I_i^{n+1} = 1 \\ Y_i^n, & \text{if } I_i^{n+1} = 0 \end{cases}$
5:       $A_i^{n+1} = \mathbf{Proj}_{\mathcal{X}_i}(Y_i^{n+1})$
6:     **end for**
7:     $n \leftarrow n + 1$
8: **until** end
---

---
**Algorithm 5:** Multi-Agent ROGD Learning under Asynchronous Feedback Loss: Stochastic Rewards
---
**Require:** Each agent $i$ picks an arbitrary $A_i^0 \in \mathbb{R}^{d_i}$
1: $n \leftarrow 0, Y_i^0 \leftarrow A_i^0$
2: **repeat**
3:     **for** each agent $i$ **do**
4:       $Y_i^{n+1} = \begin{cases} Y_i^n + \gamma_{n+1} \frac{\nabla_{a_i} R_i(A^n, \omega^{n+1})}{p_i}, & \text{if } I_i^{n+1} = 1 \\ Y_i^n, & \text{if } I_i^{n+1} = 0 \end{cases}$
5:       $A_i^{n+1} = \mathbf{Proj}_{\mathcal{X}_i}(Y_i^{n+1})$
6:     **end for**
7:     $n \leftarrow n + 1$
8: **until** end
---

---
**Algorithm 6:** Multi-Agent ROGD Learning under Asynchronous Feedback Loss: Noisy Gradient
---
**Require:** Each agent $i$ picks an arbitrary $A_i^0 \in \mathbb{R}^{d_i}$
1: $n \leftarrow 0, Y_i^0 \leftarrow A_i^0$
2: **repeat**
3:     **for** each agent $i$ **do**
4:       $Y_i^{n+1} = \begin{cases} Y_i^n + \gamma_{n+1}\left(\frac{\nabla_{a_i} r_i(A^n)}{p_i} + \xi_i^{n+1}\right), & \text{if } I_i^{n+1} = 1 \\ Y_i^n, & \text{if } I_i^{n+1} = 0 \end{cases}$
5:       $A_i^{n+1} = \mathbf{Proj}_{\mathcal{X}_i}(Y_i^{n+1})$
6:     **end for**
7:     $n \leftarrow n + 1$
8: **until** end
---

**Assumption 1.** There exists a constant $V > 0$ such that: $\forall a \in \mathcal{X}, \forall i, \|\nabla_{a_i} R_i(a, \omega)\|_2 \leq V$ for $\Pi$-almost every $\omega$.

**Assumption 2.** $\mathbb{E}[\|\nabla_{a_i} R_i(a, \omega)\|_2^2] < \infty, \forall a \in \mathcal{X}, \forall i$.

## C Algorithm and energy function

We recall $L(y) = \|a^*\|_2^2 - \|\mathbf{Proj}_{\mathcal{X}}(y)\|_2^2 + 2\langle y, \mathbf{Proj}_{\mathcal{X}}(y) - a^* \rangle$. To prove Lemma 3.2, we break it into two steps, which are stated and proved in Lemma C.1 and Lemma C.2 respectively. Further, we prove a more general case by defining $L(a, y) = \|a\|_2^2 - \|\mathbf{Proj}_{\mathcal{X}}(y)\|_2^2 + 2\langle y, \mathbf{Proj}_{\mathcal{X}}(y) - a \rangle$.

**Lemma C.1.** *For any $a \in \mathcal{X}, y \in \mathbb{R}^d$, $L(a, y) \geq 0$ with equality if and only if $\mathbf{Proj}_{\mathcal{X}}(y) = a$.*

*Proof.* Per the definition of the Lyapunov function, we have:

$$L(a,y) - \|\mathbf{Proj}_{\mathcal{X}}(y) - a\|_2^2 =$$

$$\|a\|_2^2 - \|\mathbf{Proj}_{\mathcal{X}}(y)\|_2^2 + 2\langle y, \mathbf{Proj}_{\mathcal{X}}(y) - a\rangle - \left\{\|\mathbf{Proj}_{\mathcal{X}}(y)\|_2^2 - 2\langle\mathbf{Proj}_{\mathcal{X}}(y), a\rangle + \|a\|_2^2\right\}$$

$$= -2\|\mathbf{Proj}_{\mathcal{X}}(y)\|_2^2 + 2\langle y - \mathbf{Proj}_{\mathcal{X}}(y), \mathbf{Proj}_{\mathcal{X}}(y) - a\rangle + 2\langle\mathbf{Proj}_{\mathcal{X}}(y), \mathbf{Proj}_{\mathcal{X}}(y) - a\rangle + 2\langle\mathbf{Proj}_{\mathcal{X}}(y), a\rangle$$

$$= 2\langle y - \mathbf{Proj}_{\mathcal{X}}(y), \mathbf{Proj}_{\mathcal{X}}(y) - a\rangle \geq 0,$$

<div align="right">(C.1)</div>

where the last inequality follows from Lemma A.1. Consequently, $L(a,y) - \|\mathbf{Proj}_{\mathcal{X}}(y) - a\|_2^2 = 0$. And if $L(a,y) = 0$, we must have $\|\mathbf{Proj}_{\mathcal{X}}(y) - a\|_2^2$, therefore implying $\mathbf{Proj}_{\mathcal{X}}(y) = a$. Similarly, one can also see here that if $L(a, y_n) \to 0$, then $\|\mathbf{Proj}_{\mathcal{X}}(y_n) - a\|_2^2 \to 0$, thereby implying $\mathbf{Proj}_{\mathcal{X}}(y_n) \to x$. $\qquad\square$

**Lemma C.2.** *Fix any $a \in \mathcal{X}$.*

    1. *For any two points $y, \hat{y} \in \mathbb{R}^d$:*
$$\|\mathbf{Proj}_{\mathcal{X}}(y) - \hat{y}\|_2^2 - \|\mathbf{Proj}_{\mathcal{X}}(\hat{y}) - \hat{y}\|_2^2 \leq \|y - \hat{y}\|_2^2.$$

    2. *For any $y, \Delta y \in \mathbb{R}^d$:*
$$L(a, y + \Delta y) - L(a, y) \leq 2\langle\Delta y, \mathbf{Proj}_{\mathcal{X}}(y) - a\rangle + \|\Delta y\|_2^2.$$

*Proof.* We first prove the first claim. By expanding it, we have:

$$\|\mathbf{Proj}_{\mathcal{X}}(y) - \hat{y}\|_2^2 - \|\mathbf{Proj}_{\mathcal{X}}(\hat{y}) - \hat{y}\|_2^2 = \|\mathbf{Proj}_{\mathcal{X}}(y) - y + y - \hat{y}\|_2^2 - \|\mathbf{Proj}_{\mathcal{X}}(\hat{y}) - \hat{y}\|_2^2$$

$$= \|y - \hat{y}\|_2^2 + \|\mathbf{Proj}_{\mathcal{X}}(y) - y\|_2^2 + 2\langle\mathbf{Proj}_{\mathcal{X}}(y) - y, y - \hat{y}\rangle - \|\mathbf{Proj}_{\mathcal{X}}(\hat{y}) - \hat{y}\|_2^2 \qquad \text{(C.2)}$$

$$= \|y - \hat{y}\|_2^2 - \left\{\|\mathbf{Proj}_{\mathcal{X}}(\hat{y}) - \hat{y}\|_2^2 - \|\mathbf{Proj}_{\mathcal{X}}(y) - y\|_2^2 - 2\langle y - \mathbf{Proj}_{\mathcal{X}}(y), \hat{y} - y\rangle\right\}.$$

Now define the function $f(a, y) = \|a - y\|_2^2$. It follows easily that the solution to the problem $\max_{a \in \mathcal{X}} \|a - y\|_2^2$ is $x^*(y) = \mathbf{Proj}_{\mathcal{X}}(y)$. Consequently, by Lemma A.2, $V(y) = f(x^*(y), y)$ is a differential function in $y$ and its derivative can be computed explicitly as follows:

$$\nabla V(y) = \nabla f(x^*(y), y) = 2(y - x^*(y)) = 2(y - \mathbf{Proj}_{\mathcal{X}}(y)). \qquad \text{(C.3)}$$

Futher, since for each $a \in \mathcal{X}$, $f(a, y)$ is a convex function in $y$, and taking the maximum preserves convexity, we have $V(y)$ is also a convex function in $y$. This means that

$$V(\hat{y}) - V(y) - \langle\nabla V(y), \hat{y} - y\rangle \geq 0.$$

By Equation (C.3) and that $V(y) = f(x^*(y), y) = \|\mathbf{Proj}_{\mathcal{X}}(y) - a\|_2^2$, the above equation becomes:

$$\|\mathbf{Proj}_{\mathcal{X}}(\hat{y}) - \hat{y}\|_2^2 - \|\mathbf{Proj}_{\mathcal{X}}(y) - y\|_2^2 - 2\langle y - \mathbf{Proj}_{\mathcal{X}}(y), \hat{y} - y\rangle \geq 0.$$

Consequently, Equation (A.1) then immediately yields:

$$\|\mathbf{Proj}_{\mathcal{X}}(y) - \hat{y}\|_2^2 - \|\mathbf{Proj}_{\mathcal{X}}(\hat{y}) - \hat{y}\|_2^2 \leq \|y - \hat{y}\|_2^2.$$

We now prove the second part. Expanding using the definition of the Lyapunov function, we have:

$$L(a, y + \Delta) - L(a, y) = \|a\|_2^2 - \|\mathbf{Proj}_{\mathcal{X}}(y + \Delta)\|_2^2 + 2\langle y + \Delta, \mathbf{Proj}_{\mathcal{X}}(y + \Delta) - a\rangle$$

$$- \left\{\|a\|_2^2 - \|\mathbf{Proj}_{\mathcal{X}}(y)\|_2^2 + 2\langle y, \mathbf{Proj}_{\mathcal{X}}(y) - a\rangle\right\}$$

$$= \|\mathbf{Proj}_{\mathcal{X}}(y)\|_2^2 - \|\mathbf{Proj}_{\mathcal{X}}(y + \Delta)\|_2^2 - 2\langle y, \mathbf{Proj}_{\mathcal{X}}(y) - a\rangle + 2\langle y + \Delta, \mathbf{Proj}_{\mathcal{X}}(y + \Delta) - a\rangle$$

$$= \|\mathbf{Proj}_{\mathcal{X}}(y)\|_2^2 - \|\mathbf{Proj}_{\mathcal{X}}(y + \Delta)\|_2^2 + 2\langle y, \mathbf{Proj}_{\mathcal{X}}(y + \Delta) - \mathbf{Proj}_{\mathcal{X}}(y)\rangle$$

$$+ 2\langle\Delta, \mathbf{Proj}_{\mathcal{X}}(y) - a + \mathbf{Proj}_{\mathcal{X}}(y + \Delta) - \mathbf{Proj}_{\mathcal{X}}(y)\rangle$$

$$= 2\langle\Delta, \mathbf{Proj}_{\mathcal{X}}(y) - a\rangle + 2\langle y + \Delta, \mathbf{Proj}_{\mathcal{X}}(y + \Delta) - \mathbf{Proj}_{\mathcal{X}}(y)\rangle$$

$$+ \|\mathbf{Proj}_{\mathcal{X}}(y)\|_2^2 - \|\mathbf{Proj}_{\mathcal{X}}(y + \Delta)\|_2^2$$

$$= 2\langle\Delta, \mathbf{Proj}_{\mathcal{X}}(y) - a\rangle + \|\mathbf{Proj}_{\mathcal{X}}(y) - (y + \Delta)\|_2^2 - \|\mathbf{Proj}_{\mathcal{X}}(y + \Delta) - (y + \Delta)\|_2^2$$

$$\leq 2\langle\Delta, \mathbf{Proj}_{\mathcal{X}}(y) - a\rangle + \|\Delta\|_2^2$$

<div align="right">(C.4)</div>

where the last equality follows from completing the squares and the last inequality follows from the first part of the lemma. $\qquad\square$

# D Almost sure convergence to Nash equilibria

**Lemma D.1.**
$$\sum_{t=0}^{\infty} \gamma_{n+1} \mathbb{E}[\langle \nabla_a r(A^t), a^* - A^t \rangle] < \infty. \tag{D.1}$$

*Proof.* Per the gradient update rule in ROGD, we have $Y_i^{n+1} =$
$\begin{cases} Y_i^n + \gamma_{n+1} \frac{\nabla_{a_i} r_i(A^n)}{p_i}, & \text{if } I_i^{n+1} = 1 \\ Y_i^n, & \text{if } I_i^{n+1} = 0 \end{cases}$ This implies that $\|Y_i^{n+1} - Y_i^n\|_2^2 \leq \|\gamma_{n+1} \frac{\nabla_{a_i} r_i(A^n)}{p_i}\|_2^2$, a.s.,
and consequently:

$$\|Y^{n+1} - Y^n\|_2^2 = \sum_{i=1}^N \|Y_i^{n+1} - Y_i^n\|_2^2 \leq \sum_{i=1}^N \frac{\gamma_{n+1}^2}{p_i^2} \|\nabla_{a_i} r_i(A^n)\|_2^2. \tag{D.2}$$

Since $r_i(\cdot)$ is continuously differentiable in $a_i$ (per the regularity assumption in a continuous game), and each $\mathcal{X}_i$ is a compact space, $\|\nabla_{a_i} r_i(a)\|_2^2$ is a bounded function on $\mathcal{X}$. Therefore define:

$$C_{\max} = \sup_{i \in \mathcal{N}} \max_{a \in \mathcal{X}} \|\nabla_{a_i} r_i(a)\|_2^2, \quad p = \sum_{i=1}^N \frac{1}{p_i^2}.$$

Equation (D.2) implies that (almost surely):
$$\|Y^{n+1} - Y^n\|_2^2 \leq \gamma_{n+1}^2 p C_{\max}. \tag{D.3}$$

By the second statement in Lemma C.2 and Equation (D.3), we have:
$$\begin{aligned} L(a^*, Y^{n+1}) - L(a^*, Y^n) &\leq 2\gamma_{n+1} \langle Y^{n+1} - Y^n, A^n - a^* \rangle + \gamma_{n+1}^2 \|Y^{n+1} - Y^n\|_2^2 \\ &\leq 2\gamma_{n+1} \langle Y^{n+1} - Y^n, A^n - a^* \rangle + \gamma_{n+1}^2 p C_{\max}. \end{aligned} \tag{D.4}$$

Denote by $\mathbf{1}_E$ the indicator function, which evaluates to 1 if the event $E$ happens and evaluates to 0 otherwise. We then take the expectation of both sides of Equation (D.4) and obtain:

$$\mathbb{E}[L(a^*, Y^{n+1})] - \mathbb{E}[L(a^*, Y^n)] \leq 2 \mathbb{E}[\langle Y^{n+1} - Y^n, A^n - a^* \rangle] + \gamma_{n+1}^2 p C_{\max}$$

$$= 2\mathbb{E}\Big[ \mathbb{E}[\langle Y^{n+1} - Y^n, A^n - a^* \rangle \mid Y^n] \Big] + \gamma_{n+1}^2 p C_{\max}$$

$$= 2\mathbb{E}\Big[ \mathbb{E}[\sum_{i=1}^N \langle Y_i^{n+1} - Y_i^n, A_i^n - a_i^* \rangle \mid Y^n] \Big] + \gamma_{n+1}^2 p C_{\max}$$

$$= 2\mathbb{E}\Big[ \sum_{i=1}^N \mathbb{E}[\langle Y_i^{n+1} - Y_i^n, A_i^n - a_i^* \rangle \mid Y^n] \Big] + \gamma_{n+1}^2 p C_{\max}$$

$$= 2\gamma_n \mathbb{E}\Big[ \sum_{i=1}^N \Big\{ \mathbb{E}[\langle 0, A_i^n - a_i^* \rangle \mathbf{1}_{\{I_i^{n+1}=0\}} \mid Y^n] + \mathbb{E}[\langle \gamma_{n+1} \frac{\nabla_{a_i} r_i(A^n)}{p_i}, A_i^n - a_i^* \rangle \mathbf{1}_{\{I_i^{n+1}=1\}} \mid Y^n] \Big\} \Big] + \gamma_{n+1}^2 p C_{\max}$$

$$= 2\mathbb{E}\Big[ \sum_{i=1}^N \Big\{ \langle 0, A_i^n - a_i^* \rangle \mathbb{E}[\mathbf{1}_{\{I_i^{n+1}=0\}} \mid Y^n] + \langle \gamma_{n+1} \frac{\nabla_{a_i} r_i(A^n)}{p_i}, A_i^n - a_i^* \rangle \mathbb{E}[\mathbf{1}_{\{I_i^{n+1}=1\}} \mid Y^n] \Big\} \Big] + \gamma_{n+1}^2 p C_{\max}$$

$$= 2\mathbb{E}\Big[ \sum_{i=1}^N \Big\{ \langle 0, A_i^n - a_i^* \rangle \mathbb{E}[\mathbf{1}_{\{I_i^{n+1}=0\}}] + \langle \gamma_{n+1} \frac{\nabla_{a_i} r_i(A^n)}{p_i}, A_i^n - a_i^* \rangle \mathbb{E}[\mathbf{1}_{\{I_i^{n+1}=1\}}] \Big\} \Big] + \gamma_{n+1}^2 p C_{\max}$$

$$= 2\mathbb{E}\Big[ \sum_{i=1}^N \langle \gamma_{n+1} \frac{\nabla_{a_i} r_i(A^n)}{p_i}, A_i^n - a_i^* \rangle p_i \Big] + \gamma_{n+1}^2 p C_{\max}$$

$$= 2\gamma_{n+1} \mathbb{E}\Big[ \sum_{i=1}^N \langle \nabla_{a_i} r_i(A^n), A_i^n - a_i^* \rangle \Big] + \gamma_{n+1}^2 p C_{\max}$$

$$= 2\gamma_{n+1} \mathbb{E}\Big[ \langle \nabla_a r(A^n), A^n - a^* \rangle \Big] + \gamma_{n+1}^2 p C_{\max} = -2\gamma_{n+1} \mathbb{E}\Big[ \langle \nabla_a r(A^n), a^* - A^n \rangle \Big] + \gamma_{n+1}^2 p C_{\max}, \tag{D.5}$$

where the first equality follows from the tower property, the fifth equality follows from that $A^n$ is adapted to $Y^n$, the sixth equality follows from the feedback loss process is independent of any previous iterate and the seventh equality follows from the expectation of an indicator function equals the probability of the event.

Now telescoping yields:

$$
- \mathbb{E}[L(a^*, Y^0)] \leq \mathbb{E}[L(a^*, Y^{T+1})] - \mathbb{E}[L(a^*, Y^0)] = \sum_{t=0}^{T} \mathbb{E}[L(a^*, Y^{n+1})] - \mathbb{E}[L(a^*, Y^n)]
$$

$$
\leq -2 \sum_{t=0}^{T} \gamma_{n+1} \mathbb{E}\left[ \langle \nabla_a r(A^n), a^* - A^n \rangle \right] + 2 \sum_{t=0}^{T} \gamma_{n+1}^2 p C_{\max}
$$

$$
\leq -2 \sum_{t=0}^{\infty} \gamma_{n+1} \mathbb{E}\left[ \langle \nabla_a r(A^n), a^* - A^n \rangle \right] + 2 \sum_{t=0}^{\infty} \gamma_{n+1}^2 p C_{\max},
$$

(D.6)

where the first inequality follows from that the Lyapunov function is always non-negative (Lemma C.1) and the last inequality follows since the second inequality is true for any $T$ (and we can hence let $T$ tend to $\infty$. Since the step-size is square summable, we have $2 \sum_{t=0}^{\infty} \gamma_{n+1}^2 p C_{\max} < \infty$, and

$$
-\infty > -\mathbb{E}[L(a^*, Y^0)] - 2 \sum_{t=0}^{\infty} \gamma_{n+1}^2 p C_{\max} \geq -2 \sum_{t=0}^{\infty} \gamma_{n+1} \mathbb{E}\left[ \langle \nabla_a r(A^n), a^* - A^n \rangle \right].
$$

This immediately implies $\sum_{t=0}^{\infty} \gamma_{n+1} \mathbb{E}\left[ \langle \nabla_a r(A^n), a^* - A^n \rangle \right] < \infty$, and hence the claim is established. $\square$

**Lemma D.2.** *For every $n$, there exists a constant $C > 0$ such that:*

$$
\langle \nabla_a r(A^{n+1}), a^* - A^{n+1} \rangle - \langle \nabla_a r(A^n), a^* - A^n \rangle \leq C \alpha_{n+1}
$$

(D.7)

*Proof.* The result follows from the following chain of inequalities, where all the equalities and inequalities hold almost surely:

$$
\langle \nabla_a r(A^{n+1}), a^* - A^{n+1} \rangle - \langle \nabla_a r(A^n), a^* - A^n \rangle =
$$

$$
\langle \nabla_a r(A^{n+1}) - \nabla_a r(A^n) + \nabla_a r(A^n), a^* - A^{n+1} \rangle - \langle \nabla_a r(A^n), a^* - A^n \rangle =
$$

$$
\langle \nabla_a r(A^{n+1}) - \nabla_a r(A^n), a^* - A^{n+1} \rangle + \langle \nabla_a r(A^n), A^n - A^{n+1} \rangle =
$$

$$
\leq \|\nabla_a r(A^{n+1}) - \nabla_a r(A^n)\|_2 \|a^* - A^{n+1}\|_2 + \|\nabla_a r(A^n)\|_2 \|A^n - A^{n+1}\|_2
$$

$$
\leq C_2 \|A^{n+1} - A^n\|_2 \|a^* - A^{n+1}\|_2 + C_3 \|A^n - A^{n+1}\|_2
$$

$$
= C_2 \|\mathbf{Proj}_{\mathcal{X}}(Y^{n+1}) - \mathbf{Proj}_{\mathcal{X}}(Y^n)\|_2 \|a^* - A^{n+1}\|_2 + C_3 \|\|\mathbf{Proj}_{\mathcal{X}}(Y^{n+1}) - \mathbf{Proj}_{\mathcal{X}}(Y^n)\|_2
$$

$$
= C_2 C_4 \|\mathbf{Proj}_{\mathcal{X}}(Y^{n+1}) - \mathbf{Proj}_{\mathcal{X}}(Y^n)\|_2 + C_3 \|\|\mathbf{Proj}_{\mathcal{X}}(Y^{n+1}) - \mathbf{Proj}_{\mathcal{X}}(Y^n)\|_2
$$

$$
= C_5 \|\mathbf{Proj}_{\mathcal{X}}(Y^{n+1}) - \mathbf{Proj}_{\mathcal{X}}(Y^n)\|_2 \leq C_5 \|Y^{n+1} - Y^n\|_2
$$

$$
\leq C_5 \sqrt{p C_{\max}} \gamma_{n+1},
$$

(D.8)

where the first inequality follows from Cauchy-Schwartz, the second inequality follows from that $\nabla_a r$ is Lipschitz continuous, the second-to-last equality follows from the actions space is compact, the second-to-last inequality follows from $\mathbf{Proj}_{\mathcal{X}}(\cdot)$ is a non-expansive map and the last inequality follows from Equation (D.3). The result then follows by defining $C \triangleq C_5 \sqrt{p C_{\max}}$. $\square$

**Theorem D.3.** *Let the reward functions be given from a variationally stable game. Then for any strictly positive probabilities $\{p_i\}_{i=1}^N$, ROGD converges almost surely to the set of Nash equilibria: $\lim_{n \to \infty} \mathrm{dist}(A^n, \mathcal{X}^*) = 0$ a.s., as[3] $n \to \infty$, where $A^n$ is a sequence generated from Algorithm 4.*

*Remark* 1. As a quick outline here (the details are in appendix), pick an arbitrary Nash equilibrium $a^* \in \mathcal{X}^*$. Lemma D.1 and Lemma D.2 will together ensure that $\lim_{n\to\infty} \langle \nabla_a r(A^t), a^* - A^t \rangle = 0$, a.s. Since $\langle \nabla_a r(a), a^* - a \rangle > 0$ if and only if $a \notin \mathcal{X}^*$ and $\langle \nabla_a r(a), a^* - a \rangle = 0$ if and only if $a \in \mathcal{X}^*$, it then follows by continuity of $\nabla_a r(\cdot)$ that $\lim_{n\to\infty} \langle \nabla_a r(A^t), a^* - A^t \rangle = 0$, a.s. implies $\lim_{n\to\infty} \text{dist}(A^n, \mathcal{X}^*) = 0$ a.s..

Although not mentioned in this theorem, another useful and interesting structural insight to point out here is that as $\langle \nabla_a r(A^t), a^* - A^t \rangle$ converges to 0, if it ever becomes 0 at $n$, then $A^t \in \mathcal{X}^*$, and furthermore, the joint action will stay **exactly** at that Nash equilibrium forever. Why? There are two cases to consider.

1. First, this Nash equilibrium $a^*$ is an interior point in $\mathcal{X}$. In this case, $\nabla_a r(a^*) = 0$ and hence $\nabla_a r(A^n) = 0$. Consequently, per the ROGD update rule, whether any agent updates or not does not matter: either the gradient is not received, in which case no gradient update happens; or a gradient is received, but at this Nash equilibrium, it is 0 and therefore nobody will want to make any update.

2. Second, this Nash equilibrium $a^*$ is a boundary point in $\mathcal{X}$. In this case, $\nabla_a r(a^*)$ may not be 0, but it always points outside the feasible action set $\mathcal{X}$. Consequently, even if an agent does receive a gradient and hence makes an gradient update, its action will immediately get projected back to the same point. As a result, even though the $Y^n$ variables can still change, the joint action $A^n$ will stay exactly at $\mathcal{X}$.

*Proof.* First, setting $\gamma_n = \gamma_{n+1}$, $\beta_n = \mathbb{E}[\langle \nabla_a r(A^n), a^* - A^n \rangle]$ and $X_n = \langle \nabla_a r(A^n), a^* - A^n \rangle$. Then all the sequences involved are non-negative. And by Lemma D.1 and Lemma D.2, we have: $\sum_{n=1}^\infty \gamma_n = \infty$, $\sum_{n=1}^\infty \gamma_n \beta_n < \infty$, $\mathbb{E}[X_n] \le \beta_n$, $|X_{n+1} - X_n| \le C\gamma_n$. Consequently, Lemma A.4 implies $X_n = \langle \nabla_a r(A^n), a^* - A^n \rangle \to 0$ almost surely.

Now fix any sequence $\{A^n\}_{n=0}^\infty$. We show that if $\langle \nabla_a r(A^n), a^* - A^n \rangle \to 0$, then $\text{dist}(A^n, \mathcal{X}^*) \to 0$. For suppose not, then there exists a subsequence $A^{n_k}$ such that $\text{dist}(A^{n_k}, \mathcal{X}^*) \ge \epsilon, \forall k$, for some $\epsilon > 0$. Consider the set $\mathcal{S} = \mathcal{X} - \mathcal{N}(\mathcal{X}^*, \epsilon)$, where $\mathcal{N}(\mathcal{X}^*, \epsilon)$ is an $\epsilon$-open neighborhood around $\mathcal{X}^*$. By its definition, $a \in \mathcal{X}$ is in $\mathcal{S}$ if and only if $\text{dist}(a, \mathcal{X}) \ge \epsilon$. Further, $\mathcal{S}$ is a closed and bounded set. Therefore, since $\nabla_a r(\cdot)$ is a continuous function on $\mathcal{X}$, it has a minimium $\langle \nabla_a r(a^n), a^* - a^n \rangle$ achieves a minimum value $c_{\min}$ on $\mathcal{S}$. In addition, by variational stability and the fact that $\langle \nabla_a r(a), a^* - a \rangle = 0$ if and only if $a \in \mathcal{X}^*$, we know $c_{min} > 0$. Consequently $\langle \nabla_a r(A^{n_k}), a^* - A^{n_k} \rangle \ge c_{\min}, \forall k$. This contradicts immediately $\langle \nabla_a r(A^n), a^* - A^n \rangle \to 0$. Since this argument is path-by-path, the convergence is almost sure since $\langle \nabla_a r(A^n), a^* - A^n \rangle \to 0$ almost surely. $\qquad\square$

# E   Extensions: stochastic rewards and noisy gradients

In this section, we extend the multi-agent learning under asychronous feedback loss setup to situations where agents realized rewards in each iteration are random (Section E.1). This setting, as we then establish in Section E.2, is equivalent to an equally interesting setting where the reward function is deterministic but the gradient is noisy. After discussing these two equivalent setups, we proceed to establish convergence to Nash equilibria results. We show that in bounded noise support case, we still obtain almost sure convergence to Nash for ROGD learning under asynchronous feedback loss (Section E.3). We finally look at the hardest case and establish that in the presence of both asynchronous feedback loss and unbounded noise support, ROGD learning converges to Nash equilibria in probability (Section E.4).

## E.1   Multi-Agent Learning with Stochastic Rewards

Let $(\Omega, \mathcal{F}, \Pi)$ be some underlying probability space, where $\Pi$ is the probability measure. We now consider the more general setting where agent $i$'s reward is given by:

$$r_i(a) = \mathbb{E}[R_i(a; \omega)] = \int_\Omega R_i(a; \omega) d\Pi(\omega). \tag{E.1}$$

Each agent $i$'s reward function at time $n$ is now the random quantity[4] $R_i(\cdot, \omega^{n+1})$, where $\omega^1, \omega^2, \ldots, \omega^{n+1}$ be drawn **iid** according to $\Pi$. Further, at time $t$, each agent now only observes the gradient[5] with respect to this random reward function: $\nabla R_i(A^n, \omega^{n+1})$. One important feature in this formulation is that even though the reward functions are independent across time steps (because they are drawn **iid**), they can be correlated *across agents* for the same iteration $n$: this is because we have allowed the underlying randomness $\omega$ to be drawn from a joint probability space $\Omega$ shared by all agents. Of course, this includes as a special case the setting wehre each agent $i$ has its own independent randomness for its reward function.

ROGD learning under this more general setup is now given in Algorithm 7 below:

---

**Algorithm 7:** Multi-Agent ROGD Learning under Asynchronous Feedback Loss: Stochastic Rewards

---

**Require:** Each agent $i$ picks an arbitrary $A_i^0 \in \mathbb{R}^{d_i}$
1: $n \leftarrow 0, Y_i^0 \leftarrow A_i^0$
2: **repeat**
3:     **for** each agent $i$ **do**
4:         $Y_i^{n+1} = \begin{cases} Y_i^n + \gamma_{n+1} \frac{\nabla_{a_i} R_i(A^n, \omega^{n+1})}{p_i}, & \text{if } I_i^{n+1} = 1 \\ Y_i^n, & \text{if } I_i^{n+1} = 0 \end{cases}$
5:         $A_i^{n+1} = \mathbf{Proj}_{\mathcal{X}_i}(Y_i^{n+1})$
6:     **end for**
7:     $n \leftarrow n + 1$
8: **until** end

---

We conclude this subsection with two more points.

First, variational stability is a condition on $r(a)$. In the current context, per Equation (E.1), variational stability can also be interpreted to mean that $R_i(a; \omega)$'s are variationally stable **on average**: $\sum_{i=1}^N \langle \nabla_{a_i} r_i(a), a_i - a_i^* \rangle = \sum_{i=1}^N \langle \nabla_{a_i} \mathbb{E}[R(a; \omega)], a_i - a_i^* \rangle = \mathbb{E}[\sum_{i=1}^N \langle \nabla_{a_i} R(a; \omega), a_i - a_i^* \rangle] \le 0$.

**Definition E.1.** The stochastic game $\mathcal{G} = (\mathcal{N}, \mathcal{X} = \prod_{i=1}^N \mathcal{X}_i, \{R_i(\cdot, \omega)\}_{i=1}^N)$, is mean variationally stable if its corresponding mean game $\mathcal{G} = (\mathcal{N}, \mathcal{X} = \prod_{i=1}^N \mathcal{X}_i, \{\mathbb{E}[R_i(\cdot, \omega)]\}_{i=1}^N)$ is variationally stable.

Second, two assumptions on the stochastic gradient that will be used independently later:

**Assumption 3.** There exists a constant $V > 0$ such that: $\forall a \in \mathcal{X}, \forall i, \|\nabla_{a_i} R_i(a, \omega)\|_2 \le V$ for $\Pi$-almost every $\omega$.

**Assumption 4.** $\mathbb{E}[\|\nabla_{a_i} R_i(a, \omega)\|_2^2] < \infty, \forall a \in \mathcal{X}, \forall i$.

*Remark* 2. Assumption 3 means that $\nabla_{a_i} R_i(a, \omega)$ has finite support, while Assumption 4 allows for unbounded support but only assumes bounded second moments. Of course, Assumption 3 **implies** Assumption 4 immediately.

### E.2 Equivalence to Multi-Agent Learning with Noisy Gradient

We can alternatively consider the extension of multi-agent learning under asynchronous feedback loss setting given in Algorithm 4 to the setting where the reward function $r_i(\cdot)$ is fixed and deterministic in each iteration, but the received gradient is corrupted by noise. In other words, in each iteration $n$, if agent $i$ receives a gradient, then instead of receving the exact gradient $\nabla_{a_i} r_i(A^n)$, he receives a noisy version corrupted by an additive noise. In more detail, this is given next in Algorithm 8:

**Algorithm 8:** Multi-Agent ROGD Learning under Asynchronous Feedback Loss: Noisy Gradient

---

**Require:** Each agent $i$ picks an arbitrary $A_i^0 \in \mathbb{R}^{d_i}$
1: $n \leftarrow 0, Y_i^0 \leftarrow A_i^0$
2: **repeat**
3:     **for** each agent $i$ **do**
4:         $Y_i^{n+1} = \begin{cases} Y_i^n + \gamma_{n+1}\left(\frac{\nabla_{a_i} r_i(A^n)}{p_i} + \xi_i^{n+1}\right), & \text{if } I_i^{n+1} = 1 \\ Y_i^n, & \text{if } I_i^{n+1} = 0 \end{cases}$
5:         $A_i^{n+1} = \mathbf{Proj}_{\mathcal{X}_i}(Y_i^{n+1})$
6:     **end for**
7:     $n \leftarrow n + 1$
8: **until** end

---

It turns out **iid** stochastic reward functions (as described in the previous subsection) corresponds to martingale difference noise $\xi^{n+1}$ here, as formalized in the following lemma:

**Lemma E.2.** *ROGD under stochastic rewards as given in Algorithm 7 is equivalent to ROGD under noisy gradient as given in Algorithm 8 with $\xi^{n+1} = (\xi_1^{n+1}, \xi_2^{n+1}, \ldots, \xi_N^{n+1})$ being a martingale difference sequence. Specifically, $\forall n$:*

  1. *$\xi^n$ is a martingale adapted to $Y^0, Y^1, \ldots, Y^n$.*

  2. *$\mathbb{E}[\|\xi^{n+1}\|_2] < \infty$ and $\mathbb{E}[\xi^{n+1} \mid Y^0, Y^1, \ldots, Y^n] = 0$.*

*Furthermore,*

  1. *If Assumption 3 holds, then $\|\xi^n\|_2 \leq V_*$ almost surely for all $n$ (for some positive finite constant $V_*$).*

  2. *If Assumption 4 holds, then $\mathbb{E}[\|\xi^n\|_2^2] < \infty$.*

*Remark* 3. This can be quickly seen by noting that $\xi_i^{n+1}$ in Algorithm 8 corresponds to $\frac{\nabla_{a_i} R_i(A^n, \omega^{n+1}) - \mathbb{E}[\nabla_{a_i} R_i(A^n, \omega^{n+1})]}{p_i}$ in Algorithm 7, which is zero-mean conditioned on all the past iterates. We leave all the verifcation details to appendix.

## E.3 Noise with Bounded Support: Almost Sure Convergence to Nash Equilibria

In this subsection, we work under Assumption 3 (i.e. where the noisy gradient has bounded support). It turns out that in this case, we can still get almost sure convergence to Nash equilibria. By a finer-grained analysis utilizing the martingale difference nature of the noise term $\xi^n$, we obtain the same control on the variational product sequence (as in Lemma D.1 and Lemma D.2) as given in the following lemma (the details given in the appendix):

**Lemma E.3.** *Let $A^0, A^1, \ldots, A^n$ be given from Algorithm 7 (or equivalently per Lemma 7, from Algorithm 8). Then under Assumption 3, the following two statements hold:*

  1. *$\sum_{t=0}^{\infty} \gamma_{n+1} \mathbb{E}[\langle \nabla_a r(A^t), a^* - A^t \rangle] < \infty$.*

  2. *$\langle \nabla_a r(A^{n+1}), a^* - A^{n+1} \rangle - \langle \nabla_a r(A^n), a^* - A^n \rangle \leq C\alpha_{n+1}, a.s.$*

Consequently, by the exact same reasoning as in Remark **??**, we have the following almost sure convergence result:

**Theorem E.4.** *Let the stochastic reward functions $\{R_i(a, \omega)\}_{i=1}^N$ be given from a mean variationally stable stochastic game. Then under Assumption 3, for any strictly positive probabilities $\{p_i\}_{i=1}^N$, ROGD converges almost surely to the set of Nash equilibria (of the corresponding mean game): $\lim_{n \to \infty} \mathrm{dist}(A^n, \mathcal{X}^*) = 0$ a.s., as $n \to \infty$, where $A^n$ is a sequence generated from Algorithm 7 (or from Algorithm 8).*

*Remark* 4. We mention in passing that even though almost sure convergence to Nash equilibria is still guaranteed here, there is an important difference here: if $A^n$ ever hits a Nash equilibrium, unlike in Remark **??**, it can still drift away from it due to noisy gradient. Due to space limitation, we do not expand on this point.

### E.4 Noise with Unbounded Support: Convergence to Nash Equilibria in Probability

Finally, we allow the noisy gradient to have unbounded support with only bounded second moments. In this case, the almost sure inequality in the second part of Lemma F.2 does not hold. However, it still holds in expectation, as indicated by the following lemma:

**Lemma E.5.** *Let $A^0, A^1, \ldots, A^n$ be given from Algorithm 7 (or equivalently per Lemma 7, from Algorithm 8). Then under Assumption 4, the following two statements hold:*

1. *$\sum_{t=0}^{\infty} \gamma_{n+1} \mathbb{E}[\langle \nabla_a r(A^t), a^* - A^t \rangle] < \infty$.*

2. *$\mathbb{E}[\langle \nabla_a r(A^{n+1}), a^* - A^{n+1} \rangle] - \mathbb{E}[\langle \nabla_a r(A^n), a^* - A^n \rangle] \le C\alpha_{n+1}$.*

The above then allows us to conclude $\lim_{n \to \infty} \mathbb{E}[\langle v(A^n), a^* - A^n \rangle] = 0$. And by a simple application of Markov's inequality, we obtain the convergence result in this case (see appendix for details):

**Theorem E.6.** *Let the stochastic reward functions $\{R_i(a, \omega)\}_{i=1}^N$ be given from a mean variationally stable stochastic game. Then under Assumption 4, for any strictly positive probabilities $\{p_i\}_{i=1}^N$, ROGD converges in probability to the set of Nash equilibria (of the corresponding mean game): $\forall \epsilon > 0, \lim_{n \to \infty} \mathbf{Prob}(\mathrm{dist}(A^n, \mathcal{X}^*) > \epsilon) = 0$, where $A^n$ is a sequence generated from Algorithm 7 (or from Algorithm 8).*

## F Proofs to Extensions: to stochastic rewards and noisy gradients

**Lemma F.1.** *ROGD under stochastic rewards as given in Algorithm 7 is equivalent to ROGD under noisy gradient as given in Algorithm 8 with $\xi^{n+1} = (\xi_1^{n+1}, \xi_2^{n+1}, \ldots, \xi_N^{n+1})$ being a martingale difference sequence. Specifically, for every $n$:*

1. *$\xi^n$ is a martingale adapted to $Y^0, Y^1, \ldots, Y^n$.*

2. *$\mathbb{E}[\|\xi^{n+1}\|_2] < \infty$ and $\mathbb{E}[\xi^{n+1} \mid Y^0, Y^1, \ldots, Y^n] = 0$.*

*Furthermore,*

1. *If Assumption 3 holds, then $\|\xi^n\|_2 \le V_*$ almost surely for all $n$ (for some positive finite constant $V_*$).*

2. *If Assumption 4 holds, then $\mathbb{E}[\|\xi^n\|_2^2] < \infty$.*

*Proof.* Set $\xi_i^{n+1} = \frac{1}{p_i}(\nabla_{a_i} R_i(A^n, \omega^{n+1}) - \mathbb{E}[\nabla_{a_i} R_i(A^n, \omega^{n+1})]) = \frac{1}{p_i}(\nabla_{a_i} R_i(A^n, \omega^{n+1}) - \nabla_{a_i} r_i(A^n))$ we see that the gradient update in Algorithm 8 (when the gradient is not lost) becomes:

$$Y_i^{n+1} = Y_i^n + \gamma_{n+1}\left(\frac{\nabla_{a_i} r_i(A^n)}{p_i} + \xi_i^{n+1}\right) = Y_i^n + \gamma_{n+1}\frac{\nabla_{a_i} R_i(A^n, \omega^{n+1})}{p_i}, \qquad \text{(F.1)}$$

which is exactly the gradient update in Algorithm 7.

Now it remains to check that $\xi^{n+1} = (\xi_1^{n+1}, \xi_2^{n+1}, \ldots, \xi_N^{n+1})$ is a martingale difference sequence martingale adapted to $Y^0, Y^1, \ldots, Y^{n+1}$.

First, it is easy to see that $\xi^{n+1}$ is adapted to $Y^0, Y^1, \ldots, Y^{n+1}$: each $\xi_i^{n+1}$ is determined by $Y_i^{n+1}, Y_i^n, A^t$ and since $A_i^{n+1} = \mathbf{Proj}_{\mathcal{X}_i}(Y_i^{n+1})$, $A^n$ is adapted to $Y^n$. Consequently, the sequence $Y^0, Y^1, \ldots, Y^{n+1}$ completely determines $\xi^{n+1}$. Next

$$\begin{aligned} \mathbb{E}[\|\xi^{n+1}\|_2] &= \frac{1}{p_i}\mathbb{E}\left[\left\|\nabla_{a_i} R_i(A^n, \omega^{n+1}) - \mathbb{E}[\nabla_{a_i} R_i(A^n, \omega^{n+1})]\right\|_2\right] \\ &\le \frac{1}{p_i}\left\{\mathbb{E}\left[\left\|\nabla_{a_i} R_i(A^n, \omega^{n+1})\right\|_2\right] + \left\|\mathbb{E}[\nabla_{a_i} R_i(A^n, \omega^{n+1})]\right\|_2\right\} \qquad \text{(F.2)} \\ &\le \frac{2}{p_i}\mathbb{E}[\|\nabla_{a_i} R_i(A^n, \omega^{n+1})\|_2] < \infty, \end{aligned}$$

where the second inequality follows from Jense's inequality and the last inequality follows from the fact that bounded second moments imply bounded first moments (note that under either Assumption 3 or Assumption 4, second moments are bounded). Third,

$$\mathbb{E}[\xi^{n+1} \mid Y^0, Y^1, \ldots, Y^n] = \frac{1}{p_i} \mathbb{E}[\nabla_{a_i} R_i(A^n, \omega^{n+1}) - \mathbb{E}[\nabla_{a_i} R_i(A^n, \omega^{n+1})] \mid Y^0, Y^1, \ldots, Y^n]$$

$$= \frac{1}{p_i} \left\{ \mathbb{E}[\nabla_{a_i} R_i(A^n, \omega^{n+1})] - \mathbb{E}[\nabla_{a_i} R_i(A^n, \omega^{n+1})] \right\} = 0,$$

(F.3)

where the second equation follows from $A^n$ is adapted to $Y^0, Y^1, \ldots, Y^n$ and $\omega^{n+1}$ is independent of $Y^0, Y^1, \ldots, Y^n$.

Finally, when Assumption 3 holds and using the fact that $\mathbb{E}[\nabla_{a_i} R_i(A^n, \omega^{n+1})] = \nabla_{a_i} \mathbb{E}[R_i(A^n, \omega^{n+1})] = \nabla_{a_i} r_i(a)$, we have:

$$\|\xi^{n+1}\|_2 = \frac{1}{p_i} \left\| \nabla_{a_i} R_i(A^n, \omega^{n+1}) - \mathbb{E}[\nabla_{a_i} R_i(A^n, \omega^{n+1})] \right\|_2$$

$$\leq \frac{1}{p_i} \left\{ \|\nabla_{a_i} R_i(A^n, \omega^{n+1})\|_2 + \|\mathbb{E}[\nabla_{a_i} R_i(A^n, \omega^{n+1})]\|_2 \right\} \leq \frac{1}{p_i} \left\{ V + \sup_{a \in \mathcal{X}} \|\nabla_{a_i} r_i(a)\|_2 \right\} \triangleq V_* < \infty,$$

(F.4)

where the last inequality follows from that $\nabla_{a_i} r_i(a)$ is continuous function and $\mathcal{X}$ is compact.

When Assumption 4 holds, we have

$$\mathbb{E}[\|\xi^{n+1}\|_2^2] = \frac{1}{p_i} \mathbb{E}[\left\| \nabla_{a_i} R_i(A^n, \omega^{n+1}) - \mathbb{E}[\nabla_{a_i} R_i(A^n, \omega^{n+1})] \right\|_2^2]$$

$$\leq \frac{2}{p_i} \left\{ \mathbb{E}[\left\| \nabla_{a_i} R_i(A^n, \omega^{n+1}) \right\|_2^2] + \left\| \mathbb{E}[\nabla_{a_i} R_i(A^n, \omega^{n+1})] \right\|_2^2 \right\} \leq 2 \frac{2}{p_i} \mathbb{E}[\| \nabla_{a_i} R_i(A^n, \omega^{n+1})\|_2^2] < \infty,$$

(F.5)

where the second-to-last inequality follows from Jensen's inequality. □

**Lemma F.2.** *Let $Y^0, Y^1, \ldots, Y^n$ be given from Algorithm 7 (or equivalently per Lemma 7, from Algorithm 8). Then under Assumption 4, the following two statements hold:*

1. *$\sum_{t=0}^{\infty} \gamma_{n+1} \mathbb{E}[\langle \nabla_a r(A^t), a^* - A^t \rangle] < \infty$.*

2. *$\mathbb{E}[\langle \nabla_a r(A^{n+1}), a^* - A^{n+1} \rangle] - \mathbb{E}[\langle \nabla_a r(A^n), a^* - A^n \rangle] \leq C\alpha_{n+1}$.*

*Proof.* We start by proving the first statement. Per the gradient update rule in Algorithm 8, we have $Y_i^{n+1} = \begin{cases} Y_i^n + \gamma_{n+1}(\frac{\nabla_{a_i} r_i(A^n)}{p_i} + \xi^{n+1}), & \text{if } I_i^{n+1} = 1 \\ Y_i^n, & \text{if } I_i^{n+1} = 0 \end{cases}$ This implies that $\|Y_i^{n+1} - Y_i^n\|_2^2 \leq \|\gamma_{n+1}(\frac{\nabla_{a_i} r_i(A^n)}{p_i} + \xi^{n+1})\|_2^2 \leq 2(\frac{\gamma_{n+1}^2}{p_i^2}\|\nabla_{a_i} r_i(A^n)\|_2^2 + \gamma_{n+1}^2\|\xi_i^{n+1}\|_2^2)$, a.s., and consequently:

$$\|Y^{n+1} - Y^n\|_2^2 = \sum_{i=1}^{N} \|Y_i^{n+1} - Y_i^n\|_2^2 \leq 2 \sum_{i=1}^{N} \left\{ \frac{\gamma_{n+1}^2}{p_i^2}\|\nabla_{a_i} r_i(A^n)\|_2^2 + \gamma_{n+1}^2\|\xi_i^{n+1}\|_2^2 \right\}$$

(F.6)

$$\leq 2 \sum_{i=1}^{N} \left\{ \frac{\gamma_{n+1}^2}{p_i^2}\|\nabla_{a_i} r_i(A^n)\|_2^2 + V_*^2 \gamma_{n+1}^2 \right\},$$

where the last inequality follows from Assumption 3. Since $r_i(\cdot)$ is continuously differentiable in $a_i$ (per the regularity assumption in a continuous game), and each $\mathcal{X}_i$ is a compact space, $\|\nabla_{a_i} r_i(a)\|_2^2$ is a bounded function on $\mathcal{X}$. Therefore define:

$$C_{\max} = \sup_{i \in \mathcal{N}} \max_{a \in \mathcal{X}} \|\nabla_{a_i} r_i(a)\|_2^2, \quad p = \sum_{i=1}^{N} \frac{1}{p_i^2}.$$

Equation (F.12) implies that (almost surely):

$$\|Y^{n+1} - Y^n\|_2^2 \leq 2(pC_{\max} + NV_*^2)\gamma_{n+1}^2 \triangleq C\gamma_{n+1}^2.$$

(F.7)

By the second statement in Lemma C.2 and Equation (F.13), we have:

$$L(a^*, Y^{n+1}) - L(a^*, Y^n) \leq 2\gamma_{n+1}\langle Y^{n+1} - Y^n, A^n - a^*\rangle + \gamma_{n+1}^2 \|Y^{n+1} - Y^n\|_2^2$$
$$\leq 2\gamma_{n+1}\langle Y^{n+1} - Y^n, A^n - a^*\rangle + C\gamma_{n+1}^2. \tag{F.8}$$

Again we use $\mathbf{1}_E$ as the indicator function, and take the expectation of both sides of Equation (D.4) to obtain:

$$\mathbb{E}[L(a^*, Y^{n+1})] - \mathbb{E}[L(a^*, Y^n)] \leq 2\,\mathbb{E}[\langle Y^{n+1} - Y^n, A^n - a^*\rangle] + C\gamma_{n+1}^2$$

$$= 2\mathbb{E}\Big[\,\mathbb{E}[\langle Y^{n+1} - Y^n, A^n - a^*\rangle \mid Y^n]\Big] + C\gamma_{n+1}^2 = 2\mathbb{E}\Big[\,\mathbb{E}[\sum_{i=1}^N \langle Y_i^{n+1} - Y_i^n, A_i^n - a_i^*\rangle \mid Y^n]\Big] + C\gamma_{n+1}^2$$

$$= 2\mathbb{E}\Big[\sum_{i=1}^N \mathbb{E}[\langle Y_i^{n+1} - Y_i^n, A_i^n - a_i^*\rangle \mid Y^n]\Big] + C\gamma_{n+1}^2$$

$$= 2\gamma_n\mathbb{E}\Big[\sum_{i=1}^N \Big\{ \mathbb{E}[\langle 0, A_i^n - a_i^*\rangle \mathbf{1}_{\{I_i^{n+1}=0\}} \mid Y^n] + \mathbb{E}[\langle \gamma_{n+1}(\frac{\nabla_{a_i} r_i(A^n)}{p_i} + \xi_i^{n+1}), A_i^n - a_i^*\rangle \mathbf{1}_{\{I_i^{n+1}=1\}} \mid Y^n]\Big\}\Big] + C\gamma_{n+1}^2$$

$$= 2\gamma_n\mathbb{E}\Big[\sum_{i=1}^N \Big\{ \langle 0, A_i^n - a_i^*\rangle \mathbb{E}[\mathbf{1}_{\{I_i^{n+1}=0\}} \mid Y^n] + \mathbb{E}[\langle \gamma_{n+1}(\frac{\nabla_{a_i} r_i(A^n)}{p_i} + \xi_i^{n+1}), A_i^n - a_i^*\rangle \mathbf{1}_{\{I_i^{n+1}=1\}} \mid Y^n]\Big\}\Big] + C\gamma_{n+1}^2$$

$$= 2\mathbb{E}\Big[\sum_{i=1}^N \Big\{ \mathbb{E}[\langle \gamma_{n+1}\xi_i^{n+1}, A_i^n - a_i^*\rangle \mathbf{1}_{\{I_i^{n+1}=1\}} \mid Y^n] + \langle \gamma_{n+1}\frac{\nabla_{a_i} r_i(A^n)}{p_i}, A_i^n - a_i^*\rangle \mathbb{E}[\mathbf{1}_{\{I_i^{n+1}=1\}} \mid Y^n]\Big\}\Big] + C\gamma_{n+1}^2$$

$$= 2\mathbb{E}\Big[\sum_{i=1}^N \Big\{ \mathbb{E}[\langle \gamma_{n+1}\xi_i^{n+1}, A_i^n - a_i^*\rangle \mid Y^n]\,\mathbb{E}[\mathbf{1}_{\{I_i^{n+1}=1\}}] + \langle \gamma_{n+1}\frac{\nabla_{a_i} r_i(A^n)}{p_i}, A_i^n - a_i^*\rangle \mathbb{E}[\mathbf{1}_{\{I_i^{n+1}=1\}}]\Big\}\Big] + C\gamma_{n+1}^2$$

$$= 2\mathbb{E}\Big[\sum_{i=1}^N \Big\{ p_i\gamma_{n+1}\,\mathbb{E}[\langle \xi_i^{n+1}, A_i^n - a_i^*\rangle \mid Y^n] + \langle \gamma_{n+1}\frac{\nabla_{a_i} r_i(A^n)}{p_i}, A_i^n - a_i^*\rangle p_i\Big\}\Big] + C\gamma_{n+1}^2$$

$$= 2\mathbb{E}\Big[\sum_{i=1}^N \langle \gamma_{n+1}\frac{\nabla_{a_i} r_i(A^n)}{p_i}, A_i^n - a_i^*\rangle p_i\Big] + C\gamma_{n+1}^2$$

$$= 2\gamma_{n+1}\mathbb{E}\Big[\sum_{i=1}^N \langle \nabla_{a_i} r_i(A^n), A_i^n - a_i^*\rangle\Big] + C\gamma_{n+1}^2$$

$$= 2\gamma_{n+1}\mathbb{E}\Big[\langle \nabla_a r(A^n), A^n - a^*\rangle\Big] + C\gamma_{n+1}^2 = -2\gamma_{n+1}\mathbb{E}\Big[\langle \nabla_a r(A^n), a^* - A^n\rangle\Big] + C\gamma_{n+1}^2, \tag{F.9}$$

where in the fourth-to-the-last equality, we have used the fact that $\xi^n$ is a martingale adapted the $Y^0, Y^1, \ldots, Y^n$ and hence in particular, $\mathbb{E}[\langle \xi_i^{n+1}, A_i^n - a_i^*\rangle \mid Y^n] = 0, \forall i$. Now telescoping yields:

$$-\mathbb{E}[L(a^*, Y^0)] \leq \mathbb{E}[L(a^*, Y^{T+1})] - \mathbb{E}[L(a^*, Y^0)] = \sum_{t=0}^T \mathbb{E}[L(a^*, Y^{n+1})] - \mathbb{E}[L(a^*, Y^n)]$$

$$\leq -2\sum_{t=0}^T \gamma_{n+1}\mathbb{E}\Big[\langle \nabla_a r(A^n), a^* - A^n\rangle\Big] + 2C\sum_{t=0}^T \gamma_{n+1}^2 \leq -2\sum_{t=0}^\infty \gamma_{n+1}\mathbb{E}\Big[\langle \nabla_a r(A^n), a^* - A^n\rangle\Big] + 2C\sum_{t=0}^\infty \gamma_{n+1}^2, \tag{F.10}$$

where the first inequality follows from that the Lyapunov function is always non-negative (Lemma C.1) and the last inequality follows since the second inequality is true for any $T$ (and we can hence let $T$ tend to $\infty$. Since the step-size is square summable, we have $2\sum_{t=0}^\infty C\gamma_{n+1}^2 < \infty$, and

$$-\infty > -\mathbb{E}[L(a^*, Y^0)] - 2\sum_{t=0}^\infty C\gamma_{n+1}^2 \geq -2\sum_{t=0}^\infty \gamma_{n+1}\mathbb{E}\Big[\langle \nabla_a r(A^n), a^* - A^n\rangle\Big].$$

This immediately implies $\sum_{t=0}^\infty \gamma_{n+1}\mathbb{E}\Big[\langle \nabla_a r(A^n), a^* - A^n\rangle\Big] < \infty$, and hence the claim.

Next, for the second statement, using the same chain of inequalities as in Equation (D.8) and by Equation (F.13), we have:

$$\langle \nabla_a r(A^{n+1}), a^* - A^{n+1} \rangle - \langle \nabla_a r(A^n), a^* - A^n \rangle = C_5 \| Y^{n+1} - Y^n \|_2 \le C_5 \sqrt{C} \gamma_{n+1}. \quad \text{(F.11)}$$

$\square$

**Theorem F.3.** *Let the stochastic reward functions $\{R_i(a, \omega)\}_{i=1}^N$ be given from a mean variationally stable stochastic game. Then under Assumption 3, for any strictly positive probabilities $\{p_i\}_{i=1}^N$, ROGD converges almost surely to the set of Nash equilibria (of the corresponding mean game): $\lim_{n\to\infty} \text{dist}(A^n, \mathcal{X}^*) = 0$ a.s., as $n \to \infty$, where $A^n$ is a sequence generated from Algorithm 7 (or from Algorithm 8).*

*Proof.* This follows the exact same reasoning as in the proof to Theorem D.3. $\square$

**Lemma F.4.** *Let $Y^0, Y^1, \ldots, Y^n$ be given from Algorithm 7 (or equivalently per Lemma 7, from Algorithm 8). Then under Assumption 4, the following two statements hold:*

1. $\sum_{t=0}^\infty \gamma_{n+1} \mathbb{E}[\langle \nabla_a r(A^t), a^* - A^t \rangle] < \infty$.

2. $\mathbb{E}[\langle \nabla_a r(A^{n+1}), a^* - A^{n+1} \rangle] - \mathbb{E}[\langle \nabla_a r(A^n), a^* - A^n \rangle] \le C\alpha_{n+1}$.

*Proof.* Similar to the proof to Lemma F.2, per the gradient update rule in Algorithm 8, we have

$$Y_i^{n+1} = \begin{cases} Y_i^n + \gamma_{n+1}\left(\frac{\nabla_{a_i} r_i(A^n)}{p_i} + \xi^{n+1}\right), & \text{if } I_i^{n+1} = 1 \\ Y_i^n, & \text{if } I_i^{n+1} = 0 \end{cases} \text{ and hence:}$$

$$\|Y^{n+1} - Y^n\|_2^2 = \sum_{i=1}^N \|Y_i^{n+1} - Y_i^n\|_2^2 \le 2 \sum_{i=1}^N \left\{ \frac{\gamma_{n+1}^2}{p_i^2} \|\nabla_{a_i} r_i(A^n)\|_2^2 + \gamma_{n+1}^2 \|\xi_i^{n+1}\|_2^2 \right\} \quad \text{(F.12)}$$

$$\le 2 \sum_{i=1}^N \left\{ \frac{\gamma_{n+1}^2}{p_i^2} C_{\max} + \gamma_{n+1}^2 \|\xi_i^{n+1}\|_2^2 \right\} \le 2pC_{\max}\gamma_{n+1}^2 + 2\gamma_{n+1}^2 \sum_{i=1}^N \|\xi_i^{n+1}\|_2^2.$$

Taking the expectation of both sides therefore yields Equation (F.12) implies that (almost surely):

$$\mathbb{E}[\|Y^{n+1} - Y^n\|_2^2] \le 2pC_{\max}\gamma_{n+1}^2 + 2\gamma_{n+1}^2 \sum_{i=1}^N \mathbb{E}[\|\xi_i^{n+1}\|_2^2] \le 2pC_{\max}\gamma_{n+1}^2 + 2\gamma_{n+1}^2 B \triangleq C\gamma_{n+1}^2,$$

(F.13)

where the last inequality follows from Assumption 4 (for some finite positive constant $C$). The rest of the proof then follows similarly as the proof to Lemma F.2. $\square$

**Theorem F.5.** *Let the stochastic reward functions $\{R_i(a, \omega)\}_{i=1}^N$ be given from a mean variationally stable stochastic game. Then under Assumption 4, for any strictly positive probabilities $\{p_i\}_{i=1}^N$, ROGD converges in probability to the set of Nash equilibria (of the corresponding mean game): $\forall \epsilon > 0, \lim_{n\to\infty} \textbf{Prob}(\text{dist}(A^n, \mathcal{X}^*) > \epsilon) = 0$, where $A^n$ is a sequence generated from Algorithm 7 (or from Algorithm 8).*

*Proof.* First, setting $a_n = \gamma_{n+1}, b_n = \mathbb{E}[\langle \nabla_a r(A^n), a^* - A^n \rangle]$ Lemma F.4 implies that and since all the sequences involved are non-negative, $\sum_{n=1}^\infty a_n = \infty, \sum_{n=1}^\infty a_n b_n < \infty$ and $|b_{n+1} - b_n| \le Ka_n$. Consequently, $\lim_{n\to\infty} b_n = \lim_{n\to\infty} \mathbb{E}[\langle v(A^n), a^* - A^n \rangle] = 0$ by Lemma A.3. Now, fix any $\epsilon$, by Markov's inequality, we have:

$$\textbf{Prob}(\langle \nabla_a r(A^n), a^* - A^n \rangle > \epsilon) \le \frac{\mathbb{E}[\langle \nabla_a r(A^n), a^* - A^n \rangle]}{\epsilon} \to 0,$$

when $n \to \infty$. Consequently, $\langle \nabla_a r(A^n), a^* - A^n \rangle$ converges to 0 in probability. By the same argument as in the proof to Theorem D.3 and leveraging the continuity of the reward function $r(\cdot)$, $A^n$ converges to $\mathcal{X}^*$ in probability. $\square$

## Footnotes

[1] In particular, this means that the whole function $r_t(\cdot)$ need not be observed.

[2] More specifically, it is called online gradient descent with lazy projection. There is also online gradient descent with eager projection; we do not discuss that variant due to space limitation. We also point out that we are technically doing gradient ascent as opposed to descent here, because we are working with rewards as opposed to costs. But since this point is easily understood, we do not introduce any new terminology for the same algorithm.

[3]Here the point-to-set distance is defined in the standard way: $\mathrm{dist}(A^n, \mathcal{X}^*) = \inf_{a^* \in \mathcal{X}^*} \|A^n - a^*\|_2$.

[4]Compare this to the previous setting, where agent $i$'s reward function in each iteration $n$ is $r_i(\cdot) = \mathbb{E}[R_i(\cdot; \omega^{n+1})]$.

[5]Here it is understood that the same regularity conditions be imposed on each $R_i(\cdot, \omega)$ to ensure $r_1(\cdot), \ldots, r_N(\cdot)$ form a continuous game. For instance, if we assume each $R_i(a; \omega)$ is continuous in $a$ and continuously differentiable in $a_i$ for $\Pi$-almost every $\omega$ and $\nabla_{a_i} R(a; \omega)$ is Lipschitz continuous in $a$ for $\Pi$-almost every $\omega$, then dominated convergence implies that all the regularity conditions on $r_i(a)$ also hold and one can freely exchange expectation with gradient: $\nabla_{a_i} r_i(a) = \nabla_{a_i} \mathbb{E}[R_i(a; \omega)] = \mathbb{E}[\nabla_{a_i} R_i(a; \omega)]$.