[Reviews · NeurIPS 2018]

Reviewer 1



n/a

Reviewer 2



This paper certainly seems to contain some new insights and interesting results but my main complaint is that it is so ill-placed in context and literature which makes it a tough read for both the expert and non-expert. On the one hand I think the authors miss really a lot of literature both in their introduction and related work section (see below), and on the other hand (more importantly) I think there are some recent related works that are not referred to or discussed, which make it hard for the reader to grasp what now exactly the main contribution of the work is and how much is already covered in the literature. Comments: - The references to MAL in general in the introduction are fairly weak and oversees quite a number of important publications; the authors should give credit to the right publications where these are due. Also note that the overview paper is quite outdated and more recent overviews are available, thinking of Hernandez-Leal, Tuyls & Weiss, Bloembergen et al. (JAIR), Shoham et al. (and several of the response papers in AIJ) etc. There is also a lot of literature on joint learning dynamics which is simply not referred to (see Bloembergen et al., Lanctot et al.) - Related work also need a lot more references and needs to make the connection to regret-based dynamics such as the work of Klos et al. (ECML), Lanctot et al., and e.g. NFSP etc. - Fictitious play and n-regret have been very clearly linked (see e.g. Viosat et al., Jofani et al., Cannot and Pilouras), papers that really need to be referred and discussed; there is overlap with the work in this paper and it is not discussed. - Some of the argumentation in the intro certainly makes sense, but a large part of the literature seems to be overlooked, and these works need to be discussed and given credit for. - in the intro maybe good to say you don’t consider stochastic rewards. - The reweighed online GD trick, by reweighing with probability p_i, defined as the expected value over the indicator value is not well explained. One weakness though is that it assumed the p_i’s are known (tackled in the last section though). - The definition of the energy function really lacks some intuition; it requires an effort to follow the discussion from here on due to a lack of description of the ideas of the author(s). - The main contributions in section 4 look nice, but I am unsure about all the the technicalities due to the compressed nature of writing this down (the supplementary material provides more clarity though) - Update after reading author response: I am happy with the authors response but do expect them to make the changes committed to.

Reviewer 3



The paper considers online gradient descent for multi-agent learning. For problems where the gradient feedback is available with some probability, the authors employ the reweighting technique of Exp3. The convergence of reweighted OGD to a Nash equilibria is proven for general continuous games. Extending the theoretical results for a more general class of games is a useful result, however, I would welcome some more practical motivation for the relevance of the result. The reweighting is a standard idea, but proving the convergence for the modified algorithm for multi-agent games is helpful. Minor issues: - line 161: it should be A_i instead of X_i. There are a few occurrences of `x' later as well. - line 203: convext -> convex (probably spell checking needed). - footnotes are overused; e.g., footnote 6 defines a distance used in Theorem 4.3. This is really an odd way to state a theorem.

Reviewer 4



The paper studies the problem of convergence of agents' strategies to a Nash equilibrium where the agents play repeatedly the same game and adjust their strategies their according to a simple no-regret algorithm. This is a very fundamental problem. The existing results focus convergence in the Cesaro's sense (i.e. convergence of the running average) to a correlated equilibrium. The current paper proves convergence in the standard sense, hence it is much stronger. This paper proves convergence for a certain class of games (which authors call "variationally stable games") when the agents use online gradient descent. The paper adds several bells and whistles to this basic result: Namely, the agents do not need to observe their reward in every round. The agents only observe the reward with certain probability. As long as the probability is known (at least approximately) and the probability is positive, the agents can use importance weighting to fix their learning rate. I did not check the proofs. ------ Suggestions for improvement: 1) It is puzzling the agents need to use the same step size. This fact seems to be crucial for the proof of convergence; otherwise there would be no point in importance weighting in Algorithm 3. Please give an intuitive explanation. 2) Definition 3.1: L has two arguments. You use only one. The appendix explains the issue; but that's too late and reader is only confused. Please explain your abuse of notation in the definition. 3) Definition 2. x_i --> a_i 4) Line 203: convext --> convex